# When Do Graph Foundation Models Transfer? A Data-Centric Theory

**Jiajun Zhu** [1]  **Ying Chen** [2]  **Peihao Wang** [1]  **Yixuan He** [2]  **Pan Li** [3]  **Aditya Akella** [1]  **Zhangyang Wang** [1]

## Abstract

Graph foundation models (GFMs) aim to reuse a single backbone across diverse graph domains, yet their transfer is often uneven and can exhibit negative transfer. While most prior work improves transfer through architectural or adaptation choices, we ask a data-centric question: *which properties of two graph domains determine how much a fixed representation model changes its outputs?* Using a graphon-based continuous limit for dense graphs, we show that for both set-based and message-passing tokenizations, any Lipschitz backbone admits an explicit decomposition of cross-domain output shift into (i) graph-specific finite-sample approximation terms and (ii) an intrinsic, relabeling-invariant domain discrepancy capturing structural mismatch. A key ingredient is positional-encoding (PE) stability: we establish stability guarantees for spectral PEs and highlight contrasting behaviors of eigenvector- versus subspace-based PEs. Experiments on synthetic and real graphs validate the theory and translate the decomposition into guidance for data curation in GFM transfer.

## 1. Introduction

Graph-structured data underpins applications from scientific discovery and cybersecurity to recommender systems and biology (Gilmer et al., 2017; Akoglu et al., 2015; Ying et al., 2018; Gaudelet et al., 2021). Yet practical graph learning often faces a recurring *model–data–task pipeline* problem: each new domain can differ in semantics, features, structure, and labels, requiring substantial redesign of preprocessing, architectures, and objectives. *Graph foundation models* (GFMs) (Mao et al., 2024) aim to reduce this burden by pretraining reusable backbones on large, diverse graph corpora, echoing the success of foundation models in language.

Despite growing interest, the promise of "one model for all graphs" remains far less clear in GFMs than in language, largely due to extreme *domain heterogeneity*: across domains, graphs differ in feature distributions, sizes, structural motifs, and appropriate inductive biases, often leading to *negative transfer*. Recent work tackles this challenge mainly via *model-centric* strategies—scalable training and transfer-enhancing interfaces such as MoE routing, transferable token vocabularies, and prompt-based adaptation (Sun et al., 2023; Wang et al., 2024; Xia & Huang, 2024; Liu et al.; Wang et al., 2025)—highlighting how architecture, tokenization, and adaptation mechanisms shape cross-domain transfer.

However, a core theoretical gap remains largely unaddressed: we still lack principled tools to *characterize how hard it is to transfer between two graph domains, independent of model choice*. In particular, given a fixed graph representation backbone, what properties of two graph data distributions determine how much the model's representations will change across domains? Without a notion of domain discrepancy tailored to graphs, it is difficult to (i) anticipate when a GFM will fail under domain shift, (ii) diagnose the sources of negative transfer, or (iii) justify data curation strategies intended to robustify GFM training.

To address this gap, we take a *data-centric* view and analyze representation shift using graphons as continuous limit objects for dense graphs (Lovász, 2012). Graphons provide a bridge between finite graphs and domain-level structure, allowing us to separate transfer difficulty into (i) finite-sample approximation error from sampling graphs and (ii) an intrinsic, relabeling-invariant domain discrepancy between latent generators. This view aligns with empirical evidence that many real graph datasets admit low-dimensional, domain-level structure, e.g., via mixtures of latent generators (Maskey et al., 2024; Han et al., 2022), while enabling precise domain-wise guarantees.

**Our contributions.** We develop a data-centric theory and experimental protocol for transfer in graph foundation models. (i) For set-based and message-passing tokenizations, we show that any Lipschitz backbone admits an explicit decomposition of cross-domain shift into graph-specific sampling/approximation terms and an intrinsic domain-discrepancy term. A key ingredient is spectral PE stability,

[1]University of Texas at Austin [2]Arizona State University [3]Georgia Institute of Technology. Correspondence to: Atlas Wang <atlaswang@utexas.edu>, Pan Li <panli@gatech.edu>.

*Proceedings of the 43rd International Conference on Machine Learning*, Seoul, South Korea. PMLR 306, 2026. Copyright 2026 by the author(s).

highlighting distinct behaviors of eigenvector-based and subspace-based encodings under perturbations. (ii) We validate these predictions with controlled studies of size shifts, structural perturbations, and spectral-gap effects. (iii) We translate the decomposition into practical guidance for evaluation and data curation, clarifying when larger-graph augmentation helps and when it does not.

## 2. Related Work

**Graph Generalization Theory.** A growing line of work studies generalization and transfer in graph learning through *graphon* models, which provide continuous limit objects for dense random graphs. Graphons have been used to formalize convergence/stability of graph convolutions on large random graphs (Keriven et al., 2020) and to analyze GNN transferability by relating finite graphs to underlying graphon operators (Ruiz et al., 2020; Maskey et al., 2023), with recent extensions to mixtures of graphons to model domain heterogeneity (Maskey et al., 2024). Building on this perspective, we focus on a GFM-motivated, *data-centric* view of cross-domain representation/output shift: for broad Lipschitz backbones and common tokenizations, rather than proving generalization for a specific GNN class.

**Comparison to Distributional Discrepancies.** Classical distances such as Maximum Mean Discrepancy (MMD) (Gretton et al., 2012) and the Gromov–Wasserstein (GW) distance (Mémoli, 2011) have also been used to compare graph domains (Xu et al., 2019a), but they measure different objects from our graphon discrepancy. MMD measures mismatch after mapping graphs into a chosen representation space, such as a kernel, feature map, or encoder space, and is therefore tied to that representation and its possible information loss. GW compares sampled finite graphs pairwise, so even graphs drawn from the same underlying generator can have nonzero distance due to sampling variation. In contrast, our graphon-based discrepancy compares domains directly at the level of their graph-generating mechanisms, without requiring an auxiliary embedding. It also separates finite-graph sampling error from the intrinsic mismatch between latent graphons, which is the quantity isolated as $\varepsilon_{\mathrm{gra}}$ in our decomposition.

**Graph Positional Encodings.** Positional encodings (PEs) are a core ingredient in graph Transformers and can substantially enhance message-passing GNNs. In particular, spectral PEs based on Laplacian eigenvectors are widely adopted and effective in practice (e.g., spectral attention and related designs (Kreuzer et al., 2021)). At the same time, eigenvector PEs inherit intrinsic ambiguities—sign flips and basis rotations within repeated or near-repeated eigenspaces—and can become sensitive to small graph perturbations. Recent work improves robustness by im-

posing equivariances/invariances or operating directly on eigenspaces: PEG introduces $O(p)$-equivariant processing with stability guarantees (Wang et al., 2022); Sign-Net/BasisNet achieves sign/basis invariance with universality results (Lim et al., 2022); and SPE proposes a provably stable and expressive encoding via eigenvalue-aware soft partitioning (Huang et al., 2024). Our work is complementary: rather than proposing a new PE, we connect PE stability to cross-domain transfer by showing how it enters our bounds through an explicit constant $C_{\mathrm{PE}}$.

## 3. Preliminaries

In this work, we study *graph-structure classification*, where labels depend only on the graph structure (e.g., adjacency or Laplacian) rather than external node attributes. This naturally yields a two-stage pipeline: (i) a tokenization step that maps nodes to positional embeddings (PEs) encoding structural information, and (ii) a backbone that aggregates node tokens into a graph-level prediction.

We analyze both structure-aware message-passing backbones (e.g., GCN (Kipf & Welling, 2017)) and structure-agnostic set-based backbones (e.g., DeepSets (Zaheer et al., 2017) and Graph Transformers (Yun et al., 2019)) in §3.1; the latter can still succeed since structure is injected into tokens via PE. For PEs (§3.2), we consider both eigenvector-based encodings and projector-based alternative.

### 3.1. Notations

**Graphs and graphons.** A (weighted) undirected graph is $G = (V, E)$ with $|V| = n$ and adjacency matrix $A \in \mathbb{R}^{n \times n}$ (symmetric). Throughout, we by default use the normalized graph operator $\Delta := A/n$ (so $A = n\Delta$); other shifts (e.g., $A$ or $D^{-1}A$) can be treated analogously. A graphon is a symmetric measurable function $W : [0,1]^2 \to [0,1]$ with induced bounded self-adjoint integral operator $(T_W f)(u) = \int_0^1 W(u,v)f(v)\,dv$.

**Signals and positional embeddings.** Node features are $X \in \mathbb{R}^{n \times d_x}$ (row $x_i$ for node $i$). A positional embedding (PE) is a node-wise map $t_G : [n] \to \mathbb{R}^{d_p}$ (stacked as $P \in \mathbb{R}^{n \times d_p}$). On the graphon side, a graphon signal is $x : [0,1] \to \mathbb{R}^{d_x}$ and a graphon PE is $t_W : [0,1] \to \mathbb{R}^{d_p}$.

**Message-passing Backbone.** We adopt the graph convolutional neural network (GCN), which covers a broad class of message-passing GNNs. Let $\Delta \in \mathbb{R}^{n \times n}$ be a (symmetric) graph shift operator (GSO) and let $x_\ell \in \mathbb{R}^{n \times F_\ell}$ denote the layer-$\ell$ feature map (with columns $x_\ell^j \in \mathbb{R}^n$). An GCN is specified by a collection of filters $H_\ell = (h_\ell^{jk})_{j \leq F_\ell, k \leq F_{\ell-1}}$ and a feature-mixing matrix $M_\ell = (m_\ell^{jk})_{j \leq F_\ell, k \leq F_{\ell-1}}$, to-

gether with an activation $\rho$. The layer update is

$$x_\ell^j = \rho\left(\sum_{k=1}^{F_{\ell-1}} m_\ell^{jk} \, h_\ell^{jk}(\Delta)(x_{\ell-1}^k)\right) \tag{1}$$

where $j = 1, \ldots, F_\ell$, $\ell = 1, \ldots, L$ and $h_\ell^{jk}(\Delta)$ denotes applying the (spectral) graph filter to the input signal. We write $\Phi_\Delta$ for the realized mapping on the graph with $\Delta$.

The graphon analogue replaces the GSO $\Delta$ by the graphon shift operator (WSO) $T_W$. Given a graphon $W$ and feature fields $x_\ell : [0,1] \to \mathbb{R}^{F_\ell}$ (with coordinates $x_\ell^j \in L^2([0,1])$), the WCN layer is defined by

$$x_\ell^j(u) = \rho\left(\sum_{k=1}^{F_{\ell-1}} m_\ell^{jk} \, h_\ell^{jk}(T_W)(x_{\ell-1}^k)(u)\right) \tag{2}$$

where $u \in [0,1]$. Both (1) and (2) share the same parameters $(H, M, \rho)$.

**Set Backbone.** The set backbone first converts a graph into a multiset/measure of PE-tokens and then applies a permutation-invariant map. On graphs, a standard DeepSets readout is

$$F_\theta(G) = \rho\left(\frac{1}{n}\sum_{i=1}^n \phi_\theta(t_G(i))\right), \tag{3}$$

where $\phi_\theta$ is a token encoder and $\rho$ is a predictor. On graphons, the sum becomes an integral over $[0,1]$:

$$F_\theta(W) = \rho\left(\int_0^1 \phi_\theta(t_W(u)) \, du\right). \tag{4}$$

Intuitively, PE provides an explicit graph-to-set (graph-to-measure) interface, so DeepSets can ignore the adjacency thereafter.

### 3.2. Positional Embedding on Graph(on)

**Definition 3.1** (Eigenvector positional encodings (Eig-PEs)). Let $T_W$ admit an eigendecomposition $\{(\lambda_\ell, \phi_\ell)\}_{\ell \geq 1}$ in $L^2([0,1])$, i.e. $T_W \phi_\ell = \lambda_\ell \phi_\ell$, $\|\phi_\ell\|_{L^2} = 1$. We index eigenpairs so that $\lambda_1 \leq \lambda_2 \leq \cdots \leq \lambda_n$. Fix $k \in \mathbb{N}$. Define the *graphon Eig-PE map*

$$t_W^{eig} := (\phi_1, \ldots, \phi_k) \in L^2([0,1]; \mathbb{R}^k).$$

For a finite graph $G$ equipped with graph shift operator $\Delta$ with eigenpairs $\{(\lambda_\ell, \varphi_\ell)\}_{\ell=1}^n$, indexed so that $\lambda_1 \leq \cdots \leq \lambda_n$, where $\varphi_\ell \in \mathbb{R}^n$ are orthonormal under the scaled inner product $\langle x, y\rangle_n := \frac{1}{n}x^\top y$, define the *graph Coord-PE tokens*

$$t_G^{eig} := (\varphi_1, \ldots, \varphi_k) \in \mathbb{R}^{n \times k}$$

and the induced *step* Eig-PE map on $[0,1]$,

$$St_G^{eig} := (S\varphi_1, \ldots, S\varphi_k) \in L^2([0,1]; \mathbb{R}^k).$$

**Definition 3.2** (Projector positional encodings (Proj-PEs)). Let $T_W$ admit an eigendecomposition $\{(\lambda_\ell, \phi_\ell)\}_{\ell \geq 1}$ in $L^2([0,1])$, i.e. $T_W \phi_\ell = \lambda_\ell \phi_\ell$ and $\|\phi_\ell\|_{L^2} = 1$. We index eigenpairs so that $\lambda_1 \leq \lambda_2 \leq \cdots \leq \lambda_n$. Fix $k \in \mathbb{N}$ and define the rank-$k$ orthogonal projector

$$\Pi_{W,k} : L^2([0,1]) \to L^2([0,1]), \; \Pi_{W,k}f := \sum_{\ell=1}^k \langle f, \phi_\ell\rangle_{L^2}\,\phi_\ell.$$

Equivalently, $\Pi_{W,k}$ has kernel

$$\Pi_{W,k}(u,v) := \sum_{\ell=1}^k \phi_\ell(u)\phi_\ell(v).$$

Define the graphon Proj-PE map with a $m$-dim readout function $r \in L^2([0,1]; \mathbb{R}^m)$, here $\Pi_{W,k}$ is extended to $L^2([0,1]; \mathbb{R}^m)$ channel-wise.

$$t_W^{proj} := \Pi_{W,k}r \in L^2([0,1]; \mathbb{R}^m).$$

For a finite graph operator $\Delta \in \mathbb{R}^{n \times n}$ with eigenpairs $\{(\lambda_\ell, \varphi_\ell)\}_{\ell=1}^n$, indexed so that $\lambda_1 \leq \cdots \leq \lambda_n$, where $\varphi_\ell \in \mathbb{R}^n$ are orthonormal under the scaled inner product $\langle x, y\rangle_n := \frac{1}{n}x^\top y$, fix $k \leq n$ and denote

$$U_k = (\varphi_1, \ldots, \varphi_k) \in \mathbb{R}^{n \times k}, \; \Pi_k = \frac{1}{n}U_kU_k^\top \in \mathbb{R}^{n \times n}.$$

Define the graph Proj-PE tokens with a readout matrix $R = [r_1, \ldots, r_m] \in \mathbb{R}^{n \times m}$:

$$t_G^{proj} := \Pi_k R \in \mathbb{R}^{n \times m}.$$

and the induced step Proj-PE map on $[0,1]$ with readout $R$ by

$$St_G^{proj} := S(\Pi_k R) \in L^2([0,1]; \mathbb{R}^m).$$

For the induced step-graphon $W_\Delta$, we take the default readout function as $r := SR$.

## 4. Theoretical Results

In this section, we develop a unified operator-level framework to quantify *transferability* across graphs of different sizes and domains. Our starting point is to embed a finite graph into a continuous object via an induced step-graphon, which enables size-independent comparison in a common function space. We then establish (i) explicit high-probability *graph-to-graphon* concentration rates, and (ii) *graphon-to-graphon* stability up to measure-preserving relabelings, defining an intrinsic domain discrepancy. These results culminate in an explicit decomposition of model output gaps into sampling errors and an intrinsic domain discrepancy term, providing actionable insight: improving cross-graph generalization amounts to simultaneously reducing the finite-sample approximation error and the latent graphon mismatch. Unless stated otherwise, all main proofs are provided in Appendix C.

### 4.1. Graphs are Step-Graphons

The first open question for graph foundation models is how to define the distance of graphs with different sizes. In this subsection, we claim that graphs are equivalent to step-graphon w.r.t model output, thus distance between graphs with different sizes can be defined in the same graphon space.

**Definition 4.1** (Induced step-graphon from a finite graph operator)**.** Fix $n \in \mathbb{N}$ and the partition $P_j = [(j-1)/n, j/n)$ for $j = 1, \ldots, n$. Given a finite (weighted) symmetric graph shift/operator $\Delta \in \mathbb{R}^{n \times n}$, define the induced self-adjoint step graphon (here $n\Delta$ equals adjacency matrix $A$)

$$W_\Delta(u, v) := \sum_{i=1}^{n} \sum_{j=1}^{n} n\Delta_{ij} \mathbf{1}_{P_i}(u) \mathbf{1}_{P_j}(v),$$

and the induced operator $T_{W_\Delta}$. For a node vector $x \in \mathbb{R}^n$, define the induced (step) graphon signal

$$(Sx)(u) := \sum_{j=1}^{n} x_j \mathbf{1}_{P_j}(u).$$

If $\pi$ is a permutation of nodes, we write $\pi(\Delta)$ for the permuted operator and $W_{\pi(\Delta)}$ for the induced step graphon.

The equivalence between finite graphs and induced step-graphons is immediate for backbone transformations: the discrete (neighborhood) averaging on graphs coincides with integration on the corresponding step-graphon. Likewise, the correspondence between graph PEs and induced step-graphon PEs is natural. We defer the formal statements and proofs to Appendix A.

### 4.2. Large Graphs are Closer to Graphon

In many real-world scenarios (Han et al., 2022), graphs with the same label can be viewed as generated from a shared underlying graphon. A central question for size-shift generalization is: *how does the graph–graphon discrepancy decay as the graph size $n$ increases?* We show that under evaluation sampling, larger graphs induce step-graphons that are provably closer to the underlying graphon in operator norm.

**Sampling Model.** Let $W : [0,1]^2 \rightarrow [0,1]$ be a symmetric graphon with integral operator $T_W$. Sample $u_1, \ldots, u_n \overset{iid}{\sim} \mathrm{Unif}(0,1)$ and set $A_{ij} := W(u_i, u_j)$. Let $u_{(1)} \leq \cdots \leq u_{(n)}$ be the order statistics and $\pi_u$ be the permutation satisfying $u_{\pi_u(i)} = u_{(i)}$. With the canonical partition $I_i := ((i-1)/n, i/n]$, define the step-graphon

$$W_{\pi_u(\Delta)}(x, y) := W(u_{(i)}, u_{(j)}) \qquad \text{for } (x, y) \in I_i \times I_j.$$

**Assumption 4.2** (Lipschitz graphon)**.** There exists $L < \infty$ such that for all $(x, y), (x', y') \in [0,1]^2$,

$$|W(x, y) - W(x', y')| \leq L\big(|x - x'| + |y - y'|\big).$$

**Theorem 4.3** (Operator-norm convergence)**.** *Under Assumption 4.2, there exists a universal constant $C$ such that for any $\delta \in (0, 1)$, with probability at least $1 - \delta$,*

$$\varepsilon_n = \big\|T_{W_{\pi_u(\Delta)}} - T_W\big\|_{L^2 \rightarrow L^2} \leq CL\Big(\sqrt{\tfrac{\log(2/\delta)}{n}} + \tfrac{1}{n}\Big).$$

*In particular, taking $\delta = n^{-c}$ yields $\varepsilon_n = O_p\big(\sqrt{\tfrac{\log n}{n}}\big)$.*

Theorem 4.3 formalizes our claim: *as $n$ grows, the sampled (step) graph operator approaches the underlying graphon operator*, so larger graphs provide a more faithful proxy of the graphon than smaller graphs.

### 4.3. Model Transferability Between Graphs

In §4.1, we embed any finite graph operator into a step-graphon operator, enabling comparison across sizes in a common continuous space. In §4.2, we show that the sampled-graph operator concentrates around its latent graphon operator, with discrepancy shrinking as graph size grows. Together, these results suggest viewing each observed graph as a finite-sample proxy of a domain graphon.

We then ask: *for two graphs of possibly different sizes and domains, how much can a fixed model's outputs differ?* Our approach compares graphs via their latent graphons and decomposes the output gap into a domain-mismatch term plus two sampling terms. For $G_1 \sim W_1$ and $G_2 \sim W_2$, any Lipschitz backbone with spectral PE tokens admits an informal bound of this form.

**Theorem 4.4** (Graph-to-graph model transferability, informal)**.** *Let $G_1, G_2$ be graphs sampled from graphons $W_1, W_2$. Under mild spectral-gap assumptions, the output discrepancy of a Lipschitz neural network satisfies*

$$\textit{(output gap)} \lesssim \underbrace{\varepsilon_1}_{\textit{sampling}_1} + \underbrace{\varepsilon_{\mathrm{gra}}}_{\textit{graphon mismatch}} + \underbrace{\varepsilon_2}_{\textit{sampling}_2}.$$

To formalize this decomposition, we combine two ingredients: (i) stability of the backbone to perturbations in its PE-token input, and (ii) stability of the positional encodings under operator perturbations. This yields our main graph-to-graph transferability theorem (Theorem 4.11). Since our emphasis is data-centric, we do not track Lipschitz constants in network parameters; the corresponding formal setup is deferred to Appendix B.

**Simplified notation.** To simplify the presentation, we denote by $f_\theta(W, x)$ a message-passing backbone and by $f_\theta(x)$

a set backbone, where $W$ enters only through its graphon shift operator $T_W$ (cf. (2)) and $x : [0, 1] \to \mathbb{R}^d$ is the input token/feature (in our setting, $x = t_W$ from §3.2). Under uniformly bounded parameters (mixing matrices and filters) and 1-Lipschitz nonlinearities, both maps are globally Lipschitz in the $L^2$ geometry; we use a single constant $L_\theta$ to summarize this dependence in §4.3.2. This matches the earlier notation: $f_\theta(W, x)$ corresponds to $\Phi_W(x)$ (and $\Phi_\Delta$ on graphs), while $f_\theta(x)$ corresponds to the DeepSets readout $F_\theta$.

**Proposition 4.5** (Stability of Lipschitz backbones, informal). *Let $W_1, W_2$ be two graphons with bounded self-adjoint operators $T_{W_1}, T_{W_2} : L^2([0, 1]) \to L^2([0, 1])$, and let $x_1, x_2 \in L^2([0, 1]; \mathbb{R}^d)$ be (normalized) token functions. Assume all backbone parameters (including feature-mixing matrices and spectral/message-passing filters) are uniformly bounded in operator norm and all nonlinearities are 1-Lipschitz. Then there exists a constant $L < \infty$ (depending only on these uniform bounds and the network depth) such that*

$$\|f_{\mathrm{mp}}(W_1, x_1) - f_{\mathrm{mp}}(W_2, x_2)\| \leq L\Big(\|x_1 - x_2\|_{L^2} +$$
$$\|T_{W_1} - T_{W_2}\|_{L^2 \to L^2}\Big),$$
$$\|f_{\mathrm{set}}(x_1) - f_{\mathrm{set}}(x_2)\| \leq L\,\|x_1 - x_2\|_{L^2}.$$

#### 4.3.1. STABILITY OF PE

**Assumption 4.6** (For Eig-PE). *Fix $k \in \mathbb{N}$. Let $W$ be a graphon with operator $T_W$. Let $\sigma(T_W)$ be the spectrum of $T_W$, and define $\mathrm{dist}(a, S) := \inf_{s \in S} |a - s|$ for $a \in \mathbb{R}$, $S \subseteq \mathbb{R}$. Assume the first $k$ eigenvalues are simple and separated:*

$$\gamma_\ell := \mathrm{dist}\big(\lambda_\ell, \, \sigma(T_W) \backslash \{\lambda_\ell\}\big) \geq \gamma > 2\varepsilon > 0, \, \ell = 1, \dots, k$$

*where $\varepsilon$ is the operator perturbation level.*

**Assumption 4.7** (For Proj-PE). *Fix $k, m \in \mathbb{N}$. Let $W$ be a graphon with operator $T_W$, and let $\sigma(T_W)$ denote its spectrum. Define $\mathrm{dist}(a, S) := \inf_{s \in S} |a - s|$. Assume the rank-$k$ spectral subspace is separated:*

$$\gamma_k := \min_{i \leq k} \mathrm{dist}\big(\lambda_i, \, \sigma(T_W) \backslash \{\lambda_1, \dots, \lambda_k\}\big) \geq \gamma > 2\varepsilon > 0$$

*where $\varepsilon$ denotes the operator perturbation level. Also assume the readout $r \in L^2([0, 1]; \mathbb{R}^m)$ used in $t_W^{\mathrm{proj}} = \Pi_{W,k} r$ satisfies*

$$\|r\|_{L^2([0,1]; \mathbb{R}^m)} \leq B.$$

**Lemma 4.8** (PE stability). *Fix $k \in \mathbb{N}$. Let $W, W'$ be two graphons with bounded self-adjoint operators $T_W, T_{W'}$ on $L^2([0, 1])$. Assume there exists a measure-preserving bijection $\pi : [0, 1] \to [0, 1]$ such that*

$$\|T_{W^\pi} - T_{W'}\|_{L^2 \to L^2} \leq \varepsilon, \qquad W^\pi(u, v) := W(\pi(u), \pi(v)).$$

*Then, under the corresponding PE regularity assumption (Assumption 4.6 for Eig-PE or 4.7 for Proj-PE), we have*

$$\|t_{W^\pi}^{\mathrm{PE}} - t_{W'}^{\mathrm{PE}}\|_{L^2([0,1]; \mathbb{R}^{d_{\mathrm{PE}}})} \leq C_{\mathrm{PE}}\,\varepsilon$$

*where $d_{\mathrm{PE}} = k/m$ for Eig/Proj-PE. In particular,*

$$C_{\mathrm{PE}} = \begin{cases} C_{\mathrm{eig}} := \sqrt{k} \max_{\ell \leq k} O(1/\gamma_\ell), & \text{(Eig-PE)}, \\ C_{\mathrm{proj}} := B\,O(1/\gamma_k), & \text{(Proj-PE)}. \end{cases}$$

*Remark* 4.9 (Eig-PE vs. Proj-PE). Eig-PE is a compact coordinate map $t_W^{\mathrm{eig}}(u) = (\phi_1(u), \dots, \phi_k(u))$, but its stability is gap-sensitive: small eigen-gaps $\gamma_\ell$ can induce large sign/rotation changes of eigenfunctions, making the coordinates unstable. Proj-PE instead encodes the *top-$k$ spectral subspace* via the projector $t_W^{\mathrm{proj}} = \Pi_{W,k} r$, which is basis-invariant within $\mathrm{span}\{\phi_1, \dots, \phi_k\}$ and whose stability depends only on the subspace gap $\gamma_k$. For notational convenience, we will use the same symbol $k$ to denote the PE output dimension for both variants, although the Proj-PE output dimension is $r$ as defined.

#### 4.3.2. TRANSFERABILITY

**Lemma 4.10** (Graphon transferability, simplified). *Let $W, W'$ be two graphons with associated operators $T_W, T_{W'}$. Assume there exists a measure-preserving bijection $\pi : [0, 1] \to [0, 1]$ such that*

$$\|T_{W^\pi} - T_{W'}\|_{L^2 \to L^2} \leq \varepsilon, \qquad W^\pi(u, v) := W(\pi(u), \pi(v)).$$

*Let $t_W, t_{W'}$ be PE token maps (same PE type) satisfy*

$$\|t_{W^\pi} - t_{W'}\|_{L^2} \leq C_{\mathrm{PE}}\,\varepsilon.$$

*Let $f_\theta(W, x)$ denote message-passing (MP) backbone and $f_\theta(x)$ set-based backbone with Lipschitz constant $L_\theta$. Then:*

$$\text{(Set)} \quad \|f_\theta(t_{W^\pi}) - f_\theta(t_{W'})\| \leq L_\theta\,C_{\mathrm{PE}}\,\varepsilon,$$
$$\text{(MP)} \quad \|f_\theta(W^\pi, t_{W^\pi}) - f_\theta(W', t_{W'})\| \leq L_\theta\,(1 + C_{\mathrm{PE}})\,\varepsilon.$$

**Theorem 4.11** (Graph transferability via error decomposition, simplified). *Let $G^{(1)}$ and $G^{(2)}$ be two graphs sampled from graphons $W^{(1)}$ and $W^{(2)}$, respectively, and let $W_{G^{(1)}}$ and $W_{G^{(2)}}$ denote their induced step-graphons (after suitable node permutations). Assume the following operator discrepancies are bounded:*

$$\varepsilon_1 := \|T_{W_{G^{(1)}}} - T_{W^{(1)}}\|_{L^2 \to L^2},$$
$$\varepsilon_2 := \|T_{W_{G^{(2)}}} - T_{W^{(2)}}\|_{L^2 \to L^2},$$

*and there exists a measure-preserving bijection $\pi$ such that*

$$\varepsilon_{\mathrm{gra}} := \|T_{(W^{(1)})^\pi} - T_{W^{(2)}}\|_{L^2 \to L^2}.$$

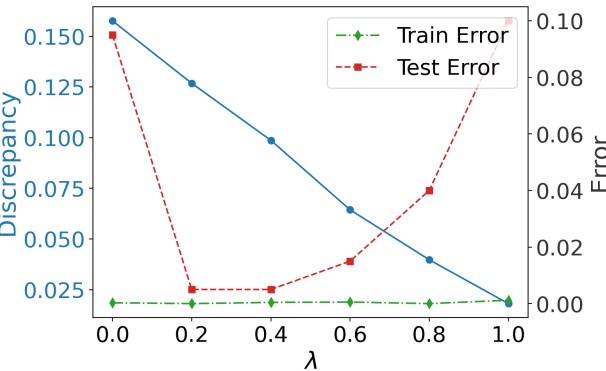

*Figure 1.* **Size shift.** As $\lambda$ increases, the train–test token-distribution discrepancy (left axis) decreases monotonically.

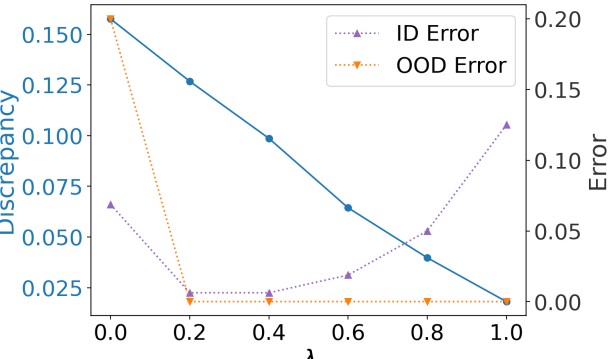

*Figure 2.* **Effect of ID error v.s. OOD error in test error.** ID error keeps decreasing while OOD error has a U-shape.

*Let $t_{G^{(1)}}, t_{G^{(2)}}$ be the (discrete) PE token maps on graphs and $t_{W^{(1)}}, t_{W^{(2)}}$ the corresponding graphon PE maps. Assume PE stability holds with a uniform constant $C_{\mathrm{PE}}$ in the sense that the induced token discrepancies satisfy*

$$\|t_{G^{(1)}} - t_{W^{(1)}}\|_{L^2} \le C_{\mathrm{PE}}\, \varepsilon_1,$$
$$\|t_{G^{(2)}} - t_{W^{(2)}}\|_{L^2} \le C_{\mathrm{PE}}\, \varepsilon_2,$$
$$\|t_{(W^{(1)})^\pi} - t_{W^{(2)}}\|_{L^2} \le C_{\mathrm{PE}}\, \varepsilon_{\mathrm{gra}}.$$

*Let $f_\theta(W, x)$ and $f_\theta(x)$ be the two backbones in the simplified notation, and assume the Lipschitz stability in Proposition 4.5 with constant $L_\theta$. Then the model output discrepancy admits the decomposition*

*(Set)* $\quad \left\| f_\theta(t_{G^{(1)}}) - f_\theta(t_{G^{(2)}}) \right\|$
$\qquad \le L_\theta\, C_{\mathrm{PE}}\, (\varepsilon_1 + \varepsilon_{\mathrm{gra}} + \varepsilon_2),$

*(MP)* $\quad \left\| f_\theta(W_{G^{(1)}}, t_{G^{(1)}}) - f_\theta(W_{G^{(2)}}, t_{G^{(2)}}) \right\|$
$\qquad \le L_\theta\, (1 + C_{\mathrm{PE}})\, (\varepsilon_1 + \varepsilon_{\mathrm{gra}} + \varepsilon_2).$

When graphs are sampled from graphons as in §4.2, the sampling terms $\varepsilon_1, \varepsilon_2$ arise from concentration of the induced step-graphon operators around the latent graphon operators and typically shrink as graph size grows. Thus, the prediction gap between two finite graphs is governed by (i) the mismatch between their underlying graphons and (ii) finite-sample approximation error controlled by graph size.

This yields two practical insights for generalizable GFMs: (a) limit the train–test *graphon distance* $\varepsilon_{\mathrm{gra}}$, since domain mismatch can dominate even for large graphs; and (b) leverage larger graphs (or aggregate data to increase effective size) to reduce $\varepsilon_1, \varepsilon_2$.

Finally, PEs enter through the stability constant $C_{\mathrm{PE}}$, which depends on the PE dimension and spectral gaps: larger dimensions or small gaps can amplify discrepancies, while projector-based encodings are typically less sensitive in small-gap regimes.

## 5. Experiments

In this section, to verify our theoretical results, we build a controlled graphon-classification benchmark to provide insights in graph foundation model training. Our experimental code and data are available at `https:github.com/VITA-Group/GraphFM`.

### 5.1. Controlled Task Settings

**Graphon classes and labels.** We fix $C$ latent graph domains represented by graphons $\{W_c\}_{c=1}^C$. To form a synthetic graph-classification task, we sample graphs from these graphons and assign each graph the label $c$ of its generating graphon. Each $W_c$ is independently designated as a train- or test-domain graphon with equal probability.

**Graph generation.** We use low-rank Fourier graphons

$$W(x, y) = \rho + \sum_{r=1}^R \lambda_r\, \phi_r(x)\phi_r(y), \qquad \rho \in (0, 1),$$

where $\{\phi_r\}$ are Fourier basis functions. To sample a size-$n$ graph, we draw latent coordinates $u_1, \ldots, u_n \overset{\text{i.i.d.}}{\sim}$ Unif$(0, 1)$ and set the (weighted) adjacency as $A_{ij} = W(u_i, u_j)$ (with $A_{ii} = 0$ and $A_{ij} = A_{ji}$ for symmetric $W$). We choose coefficients to ensure $W(x, y) \in [0, 1]$; full constructions and parameter choices are deferred to Appendix D.1.

**Train/test splits and size shift.** We instantiate $C = 8$ classes. Training graphs are sampled under a fixed total node budget (100k) across sizes $n \in \{128, 256, 512, 1024\}$, controlled by a mixing parameter $\lambda \in [0, 1]$ that reallocates budget from smaller to larger graphs as $\lambda$ increases. For evaluation, we sample a balanced test set across $n \in \{128, 256, 512, 1024, 2048\}$, where $n = 2048$ is an out-of-distribution size to probe size generalization.

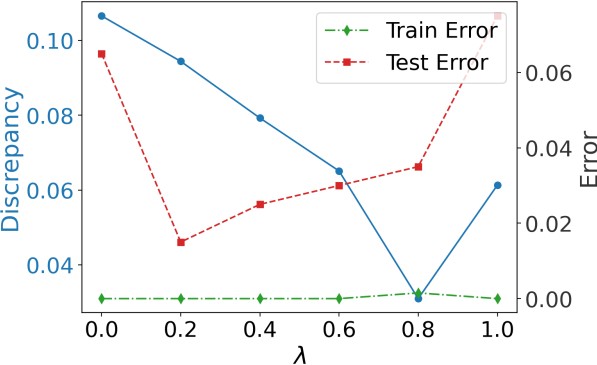

*Figure 3.* **Size shift with GIN.** The test error remains U-shaped.

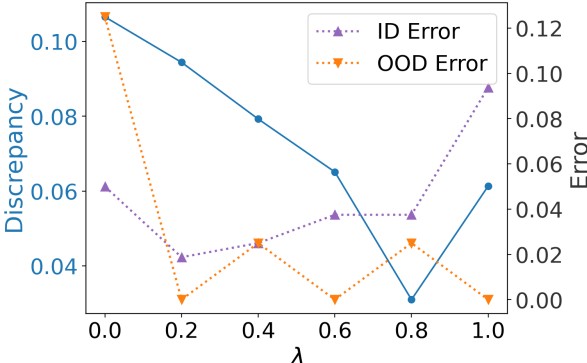

*Figure 4.* **Effect of ID error v.s. OOD error with GIN.**

**Models, training, and metrics.** Unless stated otherwise, we use a DeepSets backbone on node tokens and adopt Eig-PE by default (top-$k$ eigenvectors, with $k = 32$). To demonstrate that the data-centric terms are not tied to a specific backbone, we further include GIN (Xu et al., 2019b) as a message-passing backbone in the following results. We report test error $(1 - \mathrm{Acc})$. To diagnose distribution shift in the node-token space (including PE components), we also report a dataset-level token discrepancy between train and test, computed as a Wasserstein-type distance between the corresponding mixtures of empirical token measures. Precise estimators (subsampling, sliced Wasserstein, and all hyperparameters) are provided in Appendix D.1.

### 5.2. Size Generalization

We study how *size shift* affects generalization when training GFMs, and how graph merging interacts with this effect.

As $\lambda$ increases (Fig. 1), the train–test token-distribution discrepancy decreases monotonically, while the training error stays near zero, indicating optimization is not the bottleneck. Yet the test error exhibits a U-shape: it initially improves but degrades for large $\lambda$ even though the discrepancy continues to shrink.

We attribute this to a *coverage* issue: large $\lambda$ reallocates the training budget toward larger graphs, reducing exposure to smaller (in-domain) sizes and causing under-learning there. To verify this, we split the test error into *in-domain* sizes (seen in training) and an *out-of-domain* size ($n = 2048$). As shown in Fig. 2, the OOD error quickly saturates near zero, whereas the in-domain error increases with $\lambda$ and eventually dominates, explaining the U-shape.

The size-shift mechanism is not specific to the set-based backbone. Replacing DeepSets with a message-passing backbone, GIN (Xu et al., 2019b), reproduces the same picture (Fig. 3 and Fig. 4): as $\lambda$ increases, the train–test token discrepancy broadly decreases while the training error stays near zero, yet the test error follows the same U-shape.

The ID/OOD split also shows the same crossover as before: larger $\lambda$ improves the OOD-size error but hurts in-domain sizes. Absolute values differ slightly from Figures 1–2 because this sweep uses a freshly sampled batch of synthetic graphs.

**Graph Merging.** To improve size generalization, we study a simple *graph merging* augmentation that increases coverage of larger sizes. For each class, we estimate a class-specific latent graphon from its training graphs and then augment the training set by sampling an additional 1% graphs whose average size is $2\times$ larger than the originals. This is motivated by the view that graphs sharing a label can be treated as samples from a common generator (e.g., (Han et al., 2022)). We evaluate this strategy across different values of $\lambda$, since in practice the train–test size gap is unknown and the training set may correspond to different size mixtures.

As shown in Fig. 5, merging helps most when the train–test gap is large: at the extremes (e.g., $\lambda = 0$ or 1), a small fraction of larger synthetic graphs consistently reduces test error. For intermediate $\lambda$, where the baseline already performs well, the gains are limited and may introduce minor fluctuations, likely because augmentation perturbs an already well-aligned training distribution.

**Real-world Cases.** We evaluate graph merging on three graph-classification benchmarks (COLLAB, IMDB-BINARY, REDDIT-BINARY). Following the vanilla splits, we augment training data with class-conditioned synthetic graphs sampled from an estimated class-specific graphon, sweeping the merging ratio from 1% to 5%, and report test error (the vanilla row uses no augmentation).

As shown in Table 1, modest merging can improve generalization, but effects are dataset-dependent: COLLAB improves most at 1%, IMDB-BINARY peaks around 3% and is otherwise nearly flat, and REDDIT-BINARY shows mild gains near 3% with small fluctuations elsewhere. Overall,

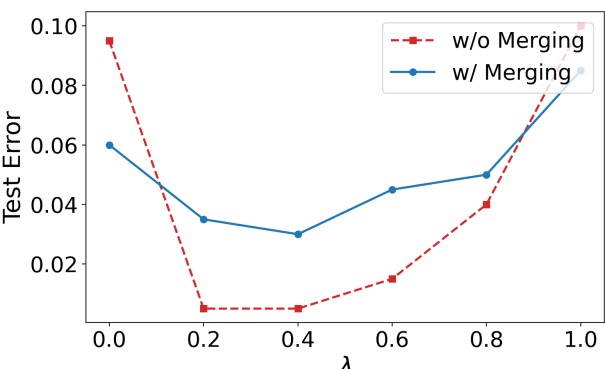

*Figure 5.* **Effect of graph merging augmentation.** We compare the test error of original training recipe and the augmented recipe with graph merging across different $\lambda$.

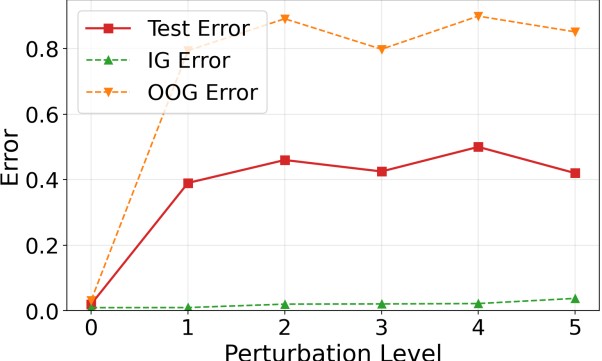

*Figure 6.* **Graphon shift.** We evaluate the model trained at $\lambda = 0.2$ while perturbing the test graphon with increasing perturbation level (larger $L_2$ distance). The training error is essentially zero.

merging helps most when the baseline has room to improve, whereas overly aggressive augmentation can hurt.

*Table 1.* **Graph merging on real-world datasets.** Test error (lower is better) when augmenting the training set with 1%–5% synthetic graphs sampled from class-specific estimated graphons. The vanilla row uses no augmentation; bold marks the best per dataset.

| Ratio | COLLAB | IMDB-BINARY | REDDIT-BINARY |
|---|---|---|---|
| Vanilla | 0.4384 | 0.4631 | 0.4108 |
| 1% | **0.4069** | 0.4631 | 0.4367 |
| 2% | 0.4428 | 0.4631 | 0.4293 |
| 3% | 0.4256 | **0.4508** | **0.4084** |
| 4% | 0.4355 | 0.5369 | 0.4182 |
| 5% | 0.4753 | 0.4631 | 0.4330 |

### 5.3. Graphon Generalization

In this subsection, we study how the train–test *graphon shift* affects generalization. We take the best-generalizing model from § 5.2, namely the model trained with the $\lambda = 0.2$ recipe (test error $< 0.05$ under the size shift setting). Keeping the training setup fixed, we then replace 50% graphs in test sets with graphs generated from *perturbed* graphons with increasing perturbation level (0–5), and report the test error together with its decomposition into an in-graphon (IG) component and an out-of-graphon (OOG) component.

As shown in Figure 6, the model remains accurate on the in-graphon samples: performance on seen graphs is largely unchanged as the perturbation level increases. In contrast, the out-of-graphon error rises dramatically as soon as we introduce a graphon perturbation, and it dominates the total test error thereafter. This suggests that *graphon shift alone* is not necessarily challenging when evaluating on familiar sizes, but *combining* graphon shift with *size extrapolation* is substantially harder and can lead to severe degradation.

### 5.4. PE's Effect on Generalization

In this subsection, we analyze how the hyperparameters and eigen-gaps affect the PE-related constants, and how these changes translate into differences in generalization performance.

**Eig-PE.** Following §5.3, we fix the size-shift setting at $\lambda = 0.2$ and keep the sampled train/test graphs unchanged, while sweeping the Laplacian Eig-PE dimension $k$. For each $k$, we retrain the same DeepSets backbone. Figure 7 reports performance together with an empirical stability proxy $\log_{10}(\sqrt{k}/\min(\text{eigengap}))$ computed from the first $k$ eigenvalues.

We observe an **expressivity–stability trade-off**. Small $k$ underfits due to limited token expressivity, while intermediate $k$ minimizes test error; for larger $k$, performance becomes less stable and degrades, especially on the OOD-size subset. This matches the gap-sensitive bound $C_{\text{eig}} \propto \sqrt{k} \max_{\ell \leq k} \gamma_\ell^{-1}$ (Lemma 4.8): as $k$ grows, shrinking eigengaps amplify the instability of eigenvector tokens under sampling noise and graph variation. Consistently, we found the train–test token discrepancy increases nearly monotonically with $k$, whereas test error is non-monotone, suggesting that early expressivity gains can initially outweigh the cost of increased cross-domain shift.

The same qualitative trade-off also appears with a message-passing backbone. Using GIN (Xu et al., 2019b), we again observe that intermediate $k$ minimizes test error, while both too-small and too-large $k$ hurt performance, with the large-$k$ degradation most visible on the OOD-size subset (Figure 9, Appendix E.1). This supports that $C_{\text{eig}}$ captures a PE-stability effect rather than a DeepSets-specific artifact.

**Proj-PE.** We also evaluate projector-style encodings that output an $m$-dimensional node token via a readout function. We test a *learnable* variant (details in Appendix D.2), where

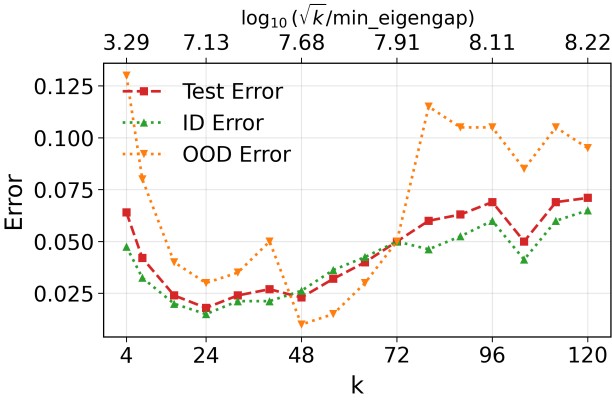

*Figure 7.* **Eig-PE sweep: test error vs chosen k.** We evaluate test error with its ID and OOD components versus the Eig-PE dimension $k$. We set $\lambda = 0.2$ and fix the train/test graph dataset. Each point averages 5 independent training runs. The top axis reports the stability proxy $\log_{10}(\sqrt{k}/\min(\text{eigengap}))$, indicating that larger $k$ corresponds to smaller minimum eigengaps and less stable tokens.

the PE module is trained jointly with the DeepSets backbone; since it converges more slowly than fixed eig-PEs, we train it for more epochs in these sweeps.

Figure 8 sweeps $k$ with a fixed readout dimension $m = 32$. The test error increases noticeably as $k$ grows, with the OOD component degrades more sharply at large $k$, suggesting that token instability still persists under Proj-PE. Compared with Eig-PE, we observe a clear degradation on the smallest in-domain graphs (e.g., $n = 128$), which can make the averaged ID error exceed the OOD error for moderate $k$; this suggests that, in our implementation, Proj-PE may be less effective on small graphs. Notably, at $k = 8$ or $16$ the OOD error is lower than under Eig-PE, suggesting that the learnable readout mechanism may improve expressivity in the low-rank regime (subject to training variance).

Experimentally, both PE types degrade as $k$ grows, and the separation predicted by Remark 4.9 is less visible in our setting. The main reason is that our experiment does not isolate this prediction: $C_{\text{proj}}$ is derived for a fixed projector, whereas our Proj-PE uses a learnable readout trained jointly with the backbone, introducing optimization confounders that make it difficult to isolate the PE stability effect alone. Moreover, the prediction applies to a specific spectral regime, where the top-$k$ eigenspace is well separated from the rest of the spectrum while the eigenvectors inside it are individually unstable. Our low-rank Fourier graphons do not produce this regime: when $k$ exceeds the intrinsic rank, the spectrum is nearly flat and provides no such well-separated top-$k$ block. A cleaner comparison would use a fixed projector together with graphs having heterogeneous gap structure, such as well-separated eigenvalue clusters with small intra-cluster gaps; we leave this comparison to future work.

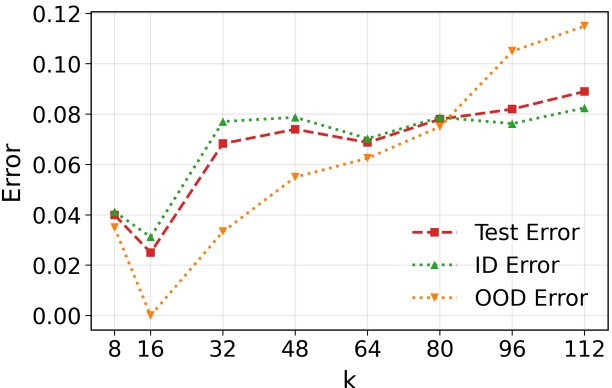

*Figure 8.* **Proj-PE sweep: test error vs chosen k (fixed m = 32).** We evaluate test error with its ID and OOD components versus the chosen spectral rank $k$, with a fixed readout dimension $m = 32$. We fix the train/test graph dataset with $\lambda = 0.2$; each point reports the average error over 5 independent training.

## 6. Conclusion

We presented a graphon-based, data-centric theory of transfer in graph foundation models, yielding a decomposition of cross-domain output shift into finite-sample approximation terms and an intrinsic, relabeling-invariant generator-mismatch term. Experiments show that allocating more training mass to larger graphs reduces token discrepancy but does not necessarily improve test error; graph merging can mitigate large gaps, generalization degrades as graphon perturbations grow, and PE exhibits an expressivity–stability trade-off as eigen-gaps shrink. Together, these results suggest that robust GFM transfer hinges on controlling generator mismatch, balancing size coverage, and choosing PE designs that avoid gap-induced instability. More broadly, our results argue that GFM transfer should be evaluated as a domain-level data problem, not only as a model-design problem. Size shifts, generator shifts, and PE-induced instability affect performance through different mechanisms; separating these factors makes negative transfer easier to diagnose and turns the theory into guidance for training-data construction.

**Limitations.** Our theory is developed for dense graphs, where graphons are the natural limiting objects and the operator-norm concentration used in Theorem 4.3 applies. Extending the analysis to sparse graphs would require replacing this step with sparse or $L^p$-graphon convergence tools (Borgs et al., 2019); we expect the same decomposition into sampling error, graphon mismatch, and PE stability to remain meaningful, but with different rates and constants. Our results also assume a Lipschitz backbone, which we empirically verify for pretrained GFMs in Appendix E.2. Extending the study to attributed, heterogeneous, or dynamic graphs remains an important direction for future work.

## Acknowledgements

ZW is supported in part by NSF Awards 2145346 (CA-REER), 2523383 (DMS), and the NSF AI Institute for Foundations of Machine Learning (IFML). AA acknowledges support from the NSF LDOS Expedition under Grant No. 2326576 and the InfraAI Center at The University of Texas at Austin. PL is partially supported by NSF grants III-2239565 and III-2428777. YH is supported by a Jetstream2 AI Fellowship and an NVIDIA Academic Grant Award. PW is in part supported by Google PhD Fellowship in Machine Learning and ML Foundations.

## Impact Statement

This paper presents work whose goal is to advance the field of Machine Learning. There are many potential societal consequences of our work, none which we feel must be specifically highlighted here.

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

# A. Proof of Graphs are Step-Graphons

**Proposition A.1** (Equivalence between graph PE and step-graphon PE). *Let $\Delta \in \mathbb{R}^{n \times n}$ and let $T_{W_\Delta}$ be its induced graphon operator on $L^2([0,1])$. Let $\{(\lambda_j, \varphi_j)\}_{j=1}^n$ be an eigendecomposition of $\Delta$. Then:*

*1. (**Eigenpair correspondence**) For every $j = 1, \ldots, n$,*

$$T_{W_\Delta} \, S\varphi_j \;=\; \lambda_j \, S\varphi_j.$$

*2. (**PE correspondence**) Fix $k \le n$. For any $u \in P_i$,*

$$(St_G)(u) \;=\; t_{W_\Delta}(u),$$

*where $t$ denotes either $t^{\mathrm{eig}}$ or $t^{\mathrm{proj}}$.*

*Proof.* **Step-operator identity.** Fix any $x \in \mathbb{R}^n$ and let $v \in P_i$. Then $W_\Delta(v,u) = n\Delta_{ij}$ for $u \in P_j$, so

$$(T_{W_\Delta} Sx)(v) = \int_0^1 W_\Delta(v,u)\,(Sx)(u)\,du = \sum_{j=1}^n \int_{P_j} n\Delta_{ij}\,x_j\,du$$

$$= \sum_{j=1}^n n\Delta_{ij}\,x_j \cdot \lambda(P_j) = \sum_{j=1}^n n\Delta_{ij}\,x_j \cdot \frac{1}{n} = \sum_{j=1}^n \Delta_{ij}x_j = (\Delta x)_i.$$

Since $(S(\Delta x))(v) = (\Delta x)_i$ for $v \in P_i$, we have the operator relation

$$T_{W_\Delta} S = S\Delta, \qquad \text{i.e.,} \qquad T_{W_\Delta}(Sx) = S(\Delta x) \;\; \forall x \in \mathbb{R}^n. \tag{5}$$

**(1) Eigenpair correspondence.** Let $\Delta\varphi_j = \lambda_j \varphi_j$. Applying (5) with $x = \varphi_j$ yields

$$T_{W_\Delta}(S\varphi_j) = S(\Delta\varphi_j) = S(\lambda_j\varphi_j) = \lambda_j\,S\varphi_j,$$

which proves (1).

**(2) Eig-PE correspondence.** Assume the discrete eigenvectors $\{\varphi_\ell\}_{\ell=1}^n$ are orthonormal under $\langle x, y\rangle_n = \frac{1}{n}x^\top y$ (equivalently $\|\varphi_\ell\|_2^2 = n$), so that $\|S\varphi_\ell\|_{L^2} = 1$ and $\{S\varphi_\ell\}$ is an $L^2$-orthonormal family. By (1), $\{(\lambda_\ell, S\varphi_\ell)\}$ is an eigendecomposition of $T_{W_\Delta}$ (up to ordering/sign), hence the graphon Eig-PE map for $W_\Delta$ can be chosen as

$$t_{W_\Delta}^{\mathrm{eig}}(u) = (S\varphi_1(u), \ldots, S\varphi_k(u)).$$

On the other hand, by Definition 3.1,

$$(St_G^{\mathrm{eig}})(u) = (S\varphi_1(u), \ldots, S\varphi_k(u)).$$

Therefore $(St_G^{\mathrm{eig}})(u) = t_{W_\Delta}^{\mathrm{eig}}(u)$ for all $u \in [0,1]$, provided the same ordering/sign convention is used.

**(3) Projector-PE correspondence.** Let $\Pi_k x = \sum_{\ell=1}^k \langle x, \varphi_\ell\rangle_n \varphi_\ell$ be the rank-$k$ projector on $\mathbb{R}^n$ (with respect to $\langle \cdot, \cdot\rangle_n$), and let $\Pi_{W_\Delta,k}$ be the rank-$k$ projector on $L^2([0,1])$ onto $\mathrm{span}\{S\varphi_1, \ldots, S\varphi_k\}$. Using $L^2$-orthonormality of $\{S\varphi_\ell\}_{\ell=1}^k$ and the identity

$$\langle Sx, S\varphi_\ell\rangle_{L^2} = \sum_{j=1}^n \int_{P_j} x_j\varphi_\ell(j)\,du = \frac{1}{n}x^\top\varphi_\ell = \langle x, \varphi_\ell\rangle_n,$$

we obtain, for any $x \in \mathbb{R}^n$,

$$\Pi_{W_\Delta,k}(Sx) = \sum_{\ell=1}^k \langle Sx, S\varphi_\ell\rangle_{L^2}\,S\varphi_\ell = \sum_{\ell=1}^k \langle x, \varphi_\ell\rangle_n\,S\varphi_\ell = S\left(\sum_{\ell=1}^k \langle x, \varphi_\ell\rangle_n\,\varphi_\ell\right) = S(\Pi_k x).$$

Take $R \in \mathbb{R}^{n \times m}$ with readout function $r := SR \in L^2([0,1]; \mathbb{R}^m)$ channel-wise, so that $r_q = (SR)_q = S(R_{:q})$. Then for each $q$,

$$t_{W_\Delta,q}^{\mathrm{proj}} = \Pi_{W_\Delta,k}r_q = \Pi_{W_\Delta,k}(S(R_{:q})) = S(\Pi_k R_{:q}) = (St_G^{\mathrm{proj}})_q.$$

Hence $t_{W_\Delta}^{\mathrm{proj}}(u) = (St_G^{\mathrm{proj}})(u)$ for all $u \in [0,1]$. This proves (3). $\square$

# B. Formal Arguments for Lipschitz Networks

**Assumption B.1** (Lipschitz DeepSets). A DeepSets-type representation on a measure $\mu$ over $\mathbb{R}^k$ is of the form

$$F_\theta(\mu) := \rho_\theta\left(\int_{\mathbb{R}^k} \phi_\theta(z)\, d\mu(z)\right),$$

where $\phi_\theta : \mathbb{R}^k \to \mathbb{R}^d$ is the token embedding and $\rho_\theta : \mathbb{R}^d \to \mathbb{R}$ is the post-aggregation map. Assume $\rho_\theta$ is $L_\rho$-Lipschitz and and $\phi_\theta$ is $L_\phi$-Lipschitz.

In particular, for an empirical measure $\mu = \frac{1}{N}\sum_{i=1}^N \delta_{z_i}$,

$$F_\theta(\mu) = \rho_\theta\left(\frac{1}{N}\sum_{i=1}^N \phi_\theta(z_i)\right),$$

which recovers the standard DeepSets architecture.

**Proposition B.2.** *Let $F_\theta$ satisfy Assumption B.1, let $\mu$ and $\nu$ be probability measures on $\mathbb{R}^k$ with finite first moment. Then*

$$\left|F_\theta(\mu) - F_\theta(\nu)\right| \le L_\rho\, L_\phi\, \mathsf{W}_1(\mu, \nu),$$

*where $\mathsf{W}_1$ is the 1-Wasserstein distance on $\mathbb{R}^k$. In particular, if $\mathsf{W}_1(\mu, \nu) \le \varepsilon$, then $|F_\theta(\mu) - F_\theta(\nu)| \le L_\rho L_\phi\, \varepsilon$.*

**Assumption B.3** (Lipschitz GNN). A graph neural network $\Phi := \Phi(H, M, \rho) : \mathbb{R}^k \to \mathbb{R}^d$ is *Lipschitz* if the following hold.

(a) **Contractive nonlinearity.** $|\rho(x) - \rho(y)| \le |x - y|$ for all $x, y \in \mathbb{R}$.

(b) **Bounded operators.** For all input graphs/graphons under consideration, the associated (self-adjoint) operators satisfy

$$\|T_W\|_{2\to 2} \le \Gamma, \qquad \|T_{W_\Delta}\|_{2\to 2} \le \Gamma.$$

(c) **Regular spectral filters.** For every layer $\ell = 1, \dots, L$ and channel pair $(j, k)$, the filter $h_\ell^{jk} \in C^1[-\Gamma, \Gamma]$, $(h_\ell^{jk})'$ is Lipschitz on $[-\Gamma, \Gamma]$, and $\|h_\ell^{jk}\|_{L^\infty[-\Gamma, \Gamma]} \le 1$.

(d) **Bounded mixing matrices.** $\|M_\ell\|_\infty \le M$ for $\ell = 1, \dots, L$ (in particular, $M = 1$ yields depth-independent constants).

**Proposition B.4** (Lipschitz stability of GNN outputs). *Let $\Phi := \Phi(H, M, \rho)$ be an $L$-layer GNN satisfying Assumption B.3. Let $W_1, W_2$ be graphons with bounded self-adjoint operators $T_{W_1}, T_{W_2} : L^2([0,1]) \to L^2([0,1])$. Let $x_1, x_2 \in L^2([0,1]; \mathbb{R}^k)$ be input feature maps satisfying $\|x_1\|_{L^2([0,1]; \mathbb{R}^k)} \le 1$ and $\|x_2\|_{L^2([0,1]; \mathbb{R}^k)} \le 1$. Then there exists a constant $C_\Phi > 0$, depending only on $\Phi$, such that*

$$\left\|\Phi_{W_1}(x_1) - \Phi_{W_2}(x_2)\right\|_{L^2([0,1]; \mathbb{R}^d)}$$
$$\le C_\Phi\left(\|T_{W_1} - T_{W_2}\|_{L^2 \to L^2} + \|x_1 - x_2\|_{L^2([0,1]; \mathbb{R}^k)}\right).$$

**Theorem B.5** (Graph-to-Graphon Stability of NNs with PE input). *Assume $\|T_{W_{\pi(\Delta)}} - T_W\|_{L^2 \to L^2} \le \varepsilon$ for some permutation $\pi$.*

(a) ***DeepSets with PE-token measures.*** *Let $F_\theta$ satisfy Assumption B.1. Let $t_G : [n] \to \mathbb{R}^k$ be a graph PE token map and $t_W : [0,1] \to \mathbb{R}^k$ a graphon PE map, and define*

$$\mu(t_G) := (S\pi t_G)_\# \lambda, \qquad \mu(t_W) := (t_W)_\# \lambda,$$

*where $\lambda$ is the uniform measure on $[0,1]$ and $S\pi t_G$ is the induced step map. Assume the corresponding PE stability bound holds as constructed in Lemma 4.8:*

$$\|S\pi t_G - t_W\|_{L^2([0,1]; \mathbb{R}^k)} \le C_{\text{PE}}\varepsilon.$$

*Then*

$$\left|F_\theta(\mu(t_G)) - F_\theta(\mu(t_W))\right| \le L_\rho L_\phi\, C_{\text{PE}}\, \varepsilon.$$

*(b) **GNN with induced step PE signals.** Let $\Phi$ satisfy Assumption B.3. Assume the corresponding PE stability bound holds as constructed in Lemma 4.8:*

$$\|S\pi t_G - t_W\|_{L^2([0,1];\mathbb{R}^k)} \leq C_{\mathrm{PE}}\varepsilon.$$

*Then there exists a constant $C > 0$ depending only on $\Phi$ and $C_{\mathrm{PE}}$ such that*

$$\left\|\Phi_{W_{\pi(\Delta)}}(S\pi t_G) - \Phi_W(t_W)\right\|_{L^2([0,1];\mathbb{R}^d)} \leq C\varepsilon.$$

## C. Main Proofs

### C.1. Proof of Theorem 4.3

We prove the high-probability bound on $\varepsilon_n = \|T_{W_{\pi_u(\Delta)}} - T_W\|_{L^2 \to L^2}$ under evaluation sampling.

**Step 1: operator norm $\leq L^2$ kernel distance.** For any $U, V \in L^2([0,1]^2)$, the difference operator $T_{U-V}$ is Hilbert–Schmidt, hence

$$\|T_U - T_V\|_{L^2 \to L^2} \leq \|U - V\|_{L^2([0,1]^2)}. \tag{6}$$

Therefore it suffices to bound $\|W_{\pi_u(\Delta)} - W\|_{L^2}$.

**Step 2: reduce to a 1D quantization error via Lipschitzness.** Define $\tau_n : [0,1] \to [0,1]$ by $\tau_n(x) := u_{(i)}$ for $x \in I_i$. Then $W_{\pi_u(\Delta)}(x,y) = W(\tau_n(x), \tau_n(y))$ for all $(x,y)$. By Lipschitzness (Assumption 4.2),

$$|W(\tau_n(x), \tau_n(y)) - W(x,y)| \leq L\big(|\tau_n(x) - x| + |\tau_n(y) - y|\big).$$

Using $(a+b)^2 \leq 2a^2 + 2b^2$ and integrating over $[0,1]^2$ gives

$$\|W_{\pi_u(\Delta)} - W\|_{L^2} \leq 2L\left(\int_0^1 |\tau_n(x) - x|^2\, dx\right)^{1/2}. \tag{7}$$

**Step 3: control $\int |\tau_n(x) - x|^2$ via DKW.** For any $x \in I_i = ((i-1)/n, i/n]$,

$$|\tau_n(x) - x| = |u_{(i)} - x| \leq |u_{(i)} - i/n| + |x - i/n| \leq |u_{(i)} - i/n| + \frac{1}{n}.$$

Thus

$$\int_0^1 |\tau_n(x) - x|^2\, dx \leq 2\max_i |u_{(i)} - i/n|^2 + \frac{2}{n^2}.$$

Let $F_n(t) = \frac{1}{n}\sum_{j=1}^n \mathbf{1}\{u_j \leq t\}$ be the empirical CDF. By the Dvoretzky–Kiefer–Wolfowitz inequality, with probability at least $1 - \delta$,

$$\sup_{t \in [0,1]} |F_n(t) - t| \leq \sqrt{\frac{\log(2/\delta)}{2n}}. \tag{8}$$

A standard consequence is

$$\max_i \left|u_{(i)} - \frac{i}{n}\right| \leq \sup_{t \in [0,1]} |F_n(t) - t| + \frac{1}{n}. \tag{9}$$

Combining (8)–(9) yields, w.p. $\geq 1 - \delta$,

$$\max_i |u_{(i)} - i/n| \leq \sqrt{\frac{\log(2/\delta)}{2n}} + \frac{1}{n}.$$

Plugging into the bound on $\int |\tau_n(x) - x|^2$ and then (7) gives, w.p. $\geq 1 - \delta$,

$$\|W_{\pi_u(\Delta)} - W\|_{L^2} \leq CL\left(\sqrt{\frac{\log(2/\delta)}{n}} + \frac{1}{n}\right)$$

for a universal constant $C$. Finally, (6) implies the same bound for $\varepsilon_n = \|T_{W_{\pi_u(\Delta)}} - T_W\|_{L^2 \to L^2}$, completing the proof. $\square$

## C.2. Proof of Corollary C.1

**Corollary C.1** (Graph-to-Graphon Stability of PE). *Fix $k \in \mathbb{N}$. Let $W$ be a graphon with bounded self-adjoint operator $T_W$ on $L^2([0,1])$. Let $\Delta$ be the graph shift operator of a finite graph $G$ sampled from $W$, and let $W_{\pi(\Delta)}$ denote the step-graphon induced by $\pi(\Delta)$. Let $t_{\pi(G)}$ be the PE map constructed from the eigenpairs of $\pi(\Delta)$. Assume*

$$\|T_{W_{\pi(\Delta)}} - T_W\|_{L^2 \to L^2} \leq \varepsilon.$$

*Then, under the corresponding PE regularity assumption (Assumption 4.6 for Eig-PE or Assumption 4.7 for Proj-PE), we have*

$$\|St^{\mathrm{PE}}_{\pi(G)} - t^{\mathrm{PE}}_W\|_{L^2([0,1];\mathbb{R}^{d_{\mathrm{PE}}})} \leq C_{\mathrm{PE}}\,\varepsilon,$$

*Proof.* (a) By Proposition A.1 (Eig-PE correspondence), after fixing the same ordering/sign convention, we may choose the first $k$ eigenfunctions of $T_{W_{\pi(\Delta)}}$ as $\{S\varphi_\ell\}_{\ell=1}^k$, where $\{(\lambda_\ell, \varphi_\ell)\}_{\ell=1}^k$ are eigenpairs of $\pi(\Delta)$. Hence

$$St^{\mathrm{eig}}_{\pi(G)} = t^{\mathrm{eig}}_{W_{\pi(\Delta)}}.$$

Under Assumption 4.6, the Davis–Kahan theorem for bounded self-adjoint operators yields eigenfunctions $\tilde{\phi}_\ell$ of $T_{W_{\pi(\Delta)}}$ (with a sign choice $\langle \tilde{\phi}_\ell, \phi_\ell \rangle_{L^2} \geq 0$) such that

$$\|\tilde{\phi}_\ell - \phi_\ell\|_{L^2([0,1])} \leq C_\ell \|T_{W_{\pi(\Delta)}} - T_W\|_{L^2 \to L^2} \leq C_\ell\,\varepsilon, \qquad C_\ell = O(1/\gamma_\ell),\ \ell = 1, \dots, k.$$

Taking $\tilde{\phi}_\ell := (t^{\mathrm{eig}}_{W_{\pi(\Delta)}})_\ell$ gives

$$\|t^{\mathrm{eig}}_{W_{\pi(\Delta)}} - t^{\mathrm{eig}}_W\|^2_{L^2([0,1];\mathbb{R}^k)} = \sum_{\ell=1}^k \|\tilde{\phi}_\ell - \phi_\ell\|^2_{L^2([0,1])} \leq k \Big( \max_{\ell \leq k} C_\ell \Big)^2 \varepsilon^2.$$

Therefore,

$$\|St^{\mathrm{eig}}_{\pi(\Delta)} - t^{\mathrm{eig}}_W\|_{L^2([0,1];\mathbb{R}^k)} = \|t^{\mathrm{eig}}_{W_{\pi(\Delta)}} - t^{\mathrm{eig}}_W\|_{L^2([0,1];\mathbb{R}^k)} \leq \sqrt{k} \Big( \max_{\ell \leq k} C_\ell \Big)\varepsilon,$$

which proves (a) with $C_{\mathrm{eig}} := \sqrt{k} \max_{\ell \leq k} C_\ell$.

(b) By Proposition A.1 (Projector-PE correspondence),

$$St^{\mathrm{proj}}_{\pi(\Delta)} = t^{\mathrm{proj}}_{W_{\pi(\Delta)}}.$$

Under Assumption 4.7, the Davis–Kahan *sin*-$\Theta$ bound yields

$$\|\Pi_{W_{\pi(\Delta)},k} - \Pi_{W,k}\|_{L^2 \to L^2} \leq C \|T_{W_{\pi(\Delta)}} - T_W\|_{L^2 \to L^2} \leq C\,\varepsilon, \qquad C = O(1/\gamma_k).$$

Thus,

$$\|St^{\mathrm{proj}}_{\pi(\Delta)} - t^{\mathrm{proj}}_W\|_{L^2} = \|t^{\mathrm{proj}}_{W_{\pi(\Delta)}} - t^{\mathrm{proj}}_W\|_{L^2} = \|(\Pi_{W_{\pi(\Delta)},k} - \Pi_{W,k})r\|_{L^2} \leq \|\Pi_{W_{\pi(\Delta)},k} - \Pi_{W,k}\|_{L^2 \to L^2} \|r\|_{L^2} \leq B\,C\,\varepsilon,$$

which proves (b) with $C_{\mathrm{proj}} := BC$. $\qquad\square$

## C.3. Proof of Lemma 4.8

*Proof.* The claim follows by the same Davis–Kahan perturbation arguments as in Lemma C.1, applied to the pair of bounded self-adjoint operators $(T_{W^\pi}, T_{W'})$. $\qquad\square$

## C.4. Proof of Proposition B.2

*Proof.* Let

$$m_\mu := \int_{\mathbb{R}^k} \phi_\theta(z)\,d\mu(z) \in \mathbb{R}^d, \qquad m_\nu := \int_{\mathbb{R}^k} \phi_\theta(z)\,d\nu(z) \in \mathbb{R}^d.$$

By $L_\rho$-Lipschitzness of $\rho_\theta$,

$$|F_\theta(\mu) - F_\theta(\nu)| \le L_\rho \|m_\mu - m_\nu\|_2.$$

Fix any coupling $\pi \in \Pi(\mu, \nu)$ of $(Z, Z') \sim \pi$. Then

$$m_\mu - m_\nu = \int_{\mathbb{R}^k \times \mathbb{R}^k} \left(\phi_\theta(z) - \phi_\theta(z')\right) d\pi(z, z').$$

Taking norms and using Jensen (or the triangle inequality for Bochner integrals),

$$\|m_\mu - m_\nu\|_2 \le \int \|\phi_\theta(z) - \phi_\theta(z')\|_2 \, d\pi(z, z') \le L_\phi \int \|z - z'\|_2 \, d\pi(z, z').$$

Taking the infimum over all couplings $\pi$ and using the primal definition of $\mathsf{W}_1$,

$$\|m_\mu - m_\nu\|_2 \le L_\phi \mathsf{W}_1(\mu, \nu).$$

Combining the above inequalities gives

$$|F_\theta(\mu) - F_\theta(\nu)| \le L_\rho L_\phi \mathsf{W}_1(\mu, \nu),$$

as claimed. $\qquad\qquad\qquad\qquad\qquad\qquad\qquad\qquad\qquad\qquad\qquad\qquad\qquad\qquad\qquad\qquad\qquad\qquad\quad \square$

### C.5. Proof of Proposition B.4

*Proof.* We follow the proof of Lemma 5.16 in (Maskey et al., 2023) (Appendix A.8) and adapt it to accommodate arbitrary discrepancies between the two inputs/graphons.

*Step 0: layerwise notation.* Let $F_0 = k$ and $F_L = d$. For $i \in \{1, 2\}$, define the layerwise feature maps $z_i^{(0)} := x_i \in L^2([0, 1]; \mathbb{R}^{F_0})$ and for $\ell = 1, \dots, L$,

$$z_i^{(\ell),j} := \rho\left(\sum_{k=1}^{F_{\ell-1}} m_\ell^{jk} \, h_\ell^{jk}(T_{W_i}) \, z_i^{(\ell-1),k}\right), \qquad j = 1, \dots, F_\ell,$$

where $M_\ell = (m_\ell^{jk})_{j,k}$ and $\rho$ is contractive (1-Lipschitz).

Let $M := \max_\ell \|M_\ell\|_\infty$. By Assumption B.3(b), $\|T_{W_i}\|_{2 \to 2} \le \Gamma$. Moreover, since each filter $h$ satisfies $h \in C^1([-\Gamma, \Gamma])$ with Lipschitz continuous derivative, the functional calculus bound (cf. (Maskey et al., 2023) Eq. (22)) yields that there exists a constant $C_h > 0$ such that for all self-adjoint operators $A, B$ with spectrum in $[-\Gamma, \Gamma]$,

$$\|h(A) - h(B)\|_{2 \to 2} \le C_h \|A - B\|_{2 \to 2}.$$

Let $C := \max_{\ell, j, k} C_{h_\ell^{jk}}$.

*Step 1: uniform bound on feature norms.* Using $\|\rho\|_{\text{Lip}} \le 1$, $\|h_\ell^{jk}(T_{W_i})\|_{2 \to 2} \le \|h_\ell^{jk}\|_\infty \le 1$, and $\|M_\ell\|_\infty \le M$, we have for each $\ell$,

$$\max_{j \le F_\ell} \|z_i^{(\ell),j}\|_{L^2} \le M \max_{k \le F_{\ell-1}} \|z_i^{(\ell-1),k}\|_{L^2} \le M^\ell \|x_i\|_{L^2}.$$

Under the normalization $\|x_i\|_{L^2} \le 1$, this gives $\max_{j \le F_\ell} \|z_i^{(\ell),j}\|_{L^2} \le M^\ell$ for $i = 1, 2$.

*Step 2: perturbation recursion.* Define $R_\ell := \max_{j \le F_\ell} \|z_1^{(\ell),j} - z_2^{(\ell),j}\|_{L^2}$. By contractivity of $\rho$ and triangle inequality, for each $\ell$,

$$
\begin{aligned}
R_\ell &\le \max_j \left\| \sum_k m_\ell^{jk} \left( h_\ell^{jk}(T_{W_1}) z_1^{(\ell-1),k} - h_\ell^{jk}(T_{W_2}) z_2^{(\ell-1),k} \right) \right\|_{L^2} \\
&\le M \max_k \| h_\ell^{jk}(T_{W_1}) z_1^{(\ell-1),k} - h_\ell^{jk}(T_{W_2}) z_2^{(\ell-1),k} \|_{L^2} \\
&\le M \max_k \left( \| (h_\ell^{jk}(T_{W_1}) - h_\ell^{jk}(T_{W_2})) z_1^{(\ell-1),k} \|_{L^2} + \| h_\ell^{jk}(T_{W_2})(z_1^{(\ell-1),k} - z_2^{(\ell-1),k}) \|_{L^2} \right) \\
&\le M \left( C \| T_{W_1} - T_{W_2} \|_{2 \to 2} \cdot \max_k \| z_1^{(\ell-1),k} \|_{L^2} + R_{\ell-1} \right).
\end{aligned}
$$

Using Step 1, $\max_k \|z_1^{(\ell-1),k}\|_{L^2} \le M^{\ell-1}$, we obtain

$$R_\ell \le MR_{\ell-1} + CM^\ell \|T_{W_1} - T_{W_2}\|_{2\to 2}.$$

Unrolling this recursion and noting $R_0 = \|x_1 - x_2\|_{L^2}$ yields

$$R_L \le M^L \|x_1 - x_2\|_{L^2} + (CL)M^L \|T_{W_1} - T_{W_2}\|_{2\to 2}.$$

*Step 3: conclude the $L^2$ output bound.* Finally,

$$\|\Phi_{W_1}(x_1) - \Phi_{W_2}(x_2)\|_{L^2([0,1];\mathbb{R}^{F_L})} \le \sqrt{F_L}\, R_L \le C_\Phi \Big( \|T_{W_1} - T_{W_2}\|_{2\to 2} + \|x_1 - x_2\|_{L^2} \Big),$$

where $C_\Phi := \sqrt{F_L}\, M^L \max\{1, CL\}$ depends only on the architecture/filter family in $\Phi$. $\qquad\square$

### C.6. Proof of Theorem B.5

*Proof of (a).* Let $\lambda$ be the uniform probability measure on $[0, 1]$. Consider the coupling $\Gamma$ between $\mu(t_G)$ and $\mu(t_W)$ defined as the law of

$$\big( S\pi t_G(U),\ t_W(U) \big) \quad \text{with } U \sim \lambda.$$

Then $\Gamma \in \Pi(\mu(t_G), \mu(t_W))$, hence by the definition of $W_1$,

$$\begin{aligned}
W_1\big(\mu(t_G), \mu(t_W)\big) &\le \int_{[0,1]} \big\| S\pi t_G(u) - t_W(u) \big\|_2 \, d\lambda(u) \\
&= \big\| S\pi t_G - t_W \big\|_{L^1([0,1];\mathbb{R}^k)} \ \le \ \big\| S\pi t_G - t_W \big\|_{L^2([0,1];\mathbb{R}^k)}.
\end{aligned}$$

By the assumed PE stability bound, $\|S\pi t_G - t_W\|_{L^2([0,1];\mathbb{R}^k)} \le C_{\mathrm{PE}}\varepsilon$, so

$$W_1\big(\mu(t_G), \mu(t_W)\big) \le C_{\mathrm{PE}}\varepsilon.$$

Finally, applying Proposition B.2 yields

$$\big| F_\theta(\mu(t_G)) - F_\theta(\mu(t_W)) \big| \ \le \ L_\rho L_\phi W_1\big(\mu(t_G), \mu(t_W)\big) \ \le \ L_\rho L_\phi C_{\mathrm{PE}}\, \varepsilon.$$

$$\square$$

*Proof of (b).* Apply Proposition B.4 to the pair

$$(W_1, x_1) = (W_{\pi(\Delta)},\ S\pi t_G), \qquad (W_2, x_2) = (W,\ t_W).$$

Then

$$\|\Phi_{W_{\pi(\Delta)}}(S\pi t_G) - \Phi_W(t_W)\|_{L^2([0,1];\mathbb{R}^d)} \le C_\Phi \Big( \|T_{W_{\pi(\Delta)}} - T_W\|_{L^2\to L^2} + \|S\pi t_G - t_W\|_{L^2([0,1];\mathbb{R}^k)} \Big).$$

Using $\|T_{W_{\pi(\Delta)}} - T_W\|_{L^2\to L^2} \le \varepsilon$ and the assumed PE stability bound $\|S\pi t_G - t_W\|_{L^2([0,1];\mathbb{R}^k)} \le C_{\mathrm{PE}}\varepsilon$, we obtain

$$\|\Phi_{W_{\pi(\Delta)}}(S\pi t_G) - \Phi_W(t_W)\|_{L^2([0,1];\mathbb{R}^d)} \le C_\Phi(1 + C_{\mathrm{PE}})\, \varepsilon.$$

Thus the claim holds with $C := C_\Phi(1 + C_{\mathrm{PE}})$. $\qquad\square$

### C.7. Proof of Lemma 4.10

**Lemma C.2** (Graphon transferability). *Let $W, W'$ be two (bounded, symmetric) graphons with bounded self-adjoint operators $T_W, T_{W'}$ on $L^2([0,1])$. Assume there exists a measure-preserving bijection $\pi : [0,1] \to [0,1]$ such that*

$$\big\|T_{W^\pi} - T_{W'}\big\|_{L^2\to L^2} \le \varepsilon, \qquad W^\pi(u,v) := W(\pi(u), \pi(v)).$$

*Assume the corresponding PE stability bound holds as constructed in Lemma 4.8:*

$$\|t_{W^\pi} - t_{W'}\|_{L^2([0,1];\mathbb{R}^k)} \le C_{\mathrm{PE}}\, \varepsilon.$$

*Then:*

*(a) **DeepSets on PE-token measures.** Let $F_\theta$ satisfy Assumption B.1, and define*

$$\mu(t_{W^\pi}) := (t_{W^\pi})_\# \lambda, \qquad \mu(t_{W'}) := (t_{W'})_\# \lambda,$$

*where $\lambda$ is the uniform probability measure on $[0,1]$. Then*

$$\left| F_\theta(\mu(t_{W^\pi})) - F_\theta(\mu(t_{W'})) \right| \leq L_\rho L_\phi C_{\mathrm{PE}}\, \varepsilon.$$

*(b) **GNN on graphons with PE signals.** Let $\Phi$ satisfy Assumption B.3. Then*

$$\left\| \Phi_{W^\pi}(t_{W^\pi}) - \Phi_{W'}(t_{W'}) \right\|_{L^2([0,1];\mathbb{R}^d)} \leq C_\Phi\, (1 + C_{\mathrm{PE}})\, \varepsilon.$$

*Proof.* The claim follows by repeating the arguments in Theorem B.5, replacing $(W_{\pi(\Delta)}, S\pi t_G)$ by $(W^\pi, t_{W^\pi})$ and $(W, t_W)$ by $(W', t_{W'})$, and using the graphon-to-graphon PE stability bound from Lemma 4.8. $\qquad\square$

## C.8. Proof of Theorem 4.11

**Theorem C.3** (Graph transferability via error decomposition). *Let $\Delta^{(1)} \in \mathbb{R}^{n_1 \times n_1}$ and $\Delta^{(2)} \in \mathbb{R}^{n_2 \times n_2}$ be two finite graph operators, sampled (possibly with different sizes) from two underlying graphons $W^{(1)}$ and $W^{(2)}$, respectively. Let $\pi_1, \pi_2$ be node permutations and consider the induced step-graphons $W_{\pi_1(\Delta^{(1)})}$ and $W_{\pi_2(\Delta^{(2)})}$ with operators $T_{W_{\pi_1(\Delta^{(1)})}}$ and $T_{W_{\pi_2(\Delta^{(2)})}}$.*

*Assume the following three discrepancies are bounded:*

$$\varepsilon_1 := \left\| T_{W_{\pi_1(\Delta^{(1)})}} - T_{W^{(1)}} \right\|_{L^2 \to L^2},$$
$$\varepsilon_2 := \left\| T_{W_{\pi_2(\Delta^{(2)})}} - T_{W^{(2)}} \right\|_{L^2 \to L^2}.$$

*and there exists a measure-preserving bijection $\pi : [0,1] \to [0,1]$ such that*

$$\varepsilon_{\mathrm{gra}} := \left\| T_{(W^{(1)})^\pi} - T_{W^{(2)}} \right\|_{L^2 \to L^2}.$$

*Let $t_{G^{(1)}}, t_{G^{(2)}}$ be the (discrete) graph PE token maps, and $t_{W^{(1)}}, t_{W^{(2)}}$ the corresponding graphon PE maps (same PE type throughout). Assume the PE stability bounds in Lemma 4.8 hold with constants $C_{\mathrm{PE}}^{(1)}$ and $C_{\mathrm{PE}}^{(2)}$, and define*

$$C_{\mathrm{PE}} := \max\left\{ C_{\mathrm{PE}}^{(1)}, C_{\mathrm{PE}}^{(2)} \right\}.$$

*Namely,*

$$\| S\pi_1 t_{G^{(1)}} - t_{W^{(1)}} \|_{L^2} \leq C_{\mathrm{PE}}\, \varepsilon_1,$$
$$\| S\pi_2 t_{G^{(2)}} - t_{W^{(2)}} \|_{L^2} \leq C_{\mathrm{PE}}\, \varepsilon_2,$$

*and*

$$\| t_{(W^{(1)})^\pi} - t_{W^{(2)}} \|_{L^2} \leq C_{\mathrm{PE}}\, \varepsilon_{\mathrm{gra}}.$$

*Then:*

*(a) **DeepSets discrepancy decomposition.** Let $F_\theta$ satisfy Assumption B.1, and define*

$$\mu^{(1)} := (S\pi_1 t_{G^{(1)}})_\# \lambda, \qquad \mu^{(2)} := (S\pi_2 t_{G^{(2)}})_\# \lambda.$$

*Then*

$$\left| F_\theta(\mu^{(1)}) - F_\theta(\mu^{(2)}) \right| \leq L_\rho L_\phi C_{\mathrm{PE}} \left( \varepsilon_1 + \varepsilon_{\mathrm{gra}} + \varepsilon_2 \right).$$

*(b) **GNN discrepancy decomposition.** Let $\Phi$ satisfy Assumption B.3. Then*

$$\left\| \Phi_{W_{\pi_1(\Delta^{(1)})}}\left(S\pi_1 t_{G^{(1)}}\right) \circ \pi - \Phi_{W_{\pi_2(\Delta^{(2)})}}\left(S\pi_2 t_{G^{(2)}}\right) \right\|_{L^2([0,1];\mathbb{R}^d)}$$
$$\leq \quad C_\Phi\, (1 + C_{\mathrm{PE}}) \left( \varepsilon_1 + \varepsilon_{\mathrm{gra}} + \varepsilon_2 \right).$$

*Proof.* Both parts follow from a triangle inequality with two applications of Theorem B.5 (graph-to-graphon) and one application of Lemma 4.10 (graphon-to-graphon), using the three discrepancies $\varepsilon_1, \varepsilon_{\mathrm{gra}}, \varepsilon_2$. $\qquad\square$

# D. Implementation Details

## D.1. Controlled Task Settings

**Fourier graphons and boundedness.** We use Fourier basis functions $\phi_{m,c}(x) = \sqrt{2}\cos(2\pi mx)$ and $\phi_{m,s}(x) = \sqrt{2}\sin(2\pi mx)$, hence $|\phi_r(x)| \leq \sqrt{2}$ and

$$\left| \sum_{r=1}^{R} \lambda_r \, \phi_r(x)\phi_r(y) \right| \leq 2 \sum_{r=1}^{R} |\lambda_r|.$$

A sufficient condition for $W(x, y) \in [0, 1]$ for all $x, y$ is

$$2 \sum_{r=1}^{R} |\lambda_r| \leq \min(\rho, 1 - \rho), \quad \text{equivalently} \quad \sum_{r=1}^{R} |\lambda_r| \leq \tfrac{1}{2} \min(\rho, 1 - \rho).$$

We set $\rho = 0.5$ and use $R = 8$ Fourier terms, with coefficients $\lambda_r$ sampled from $[-0.25, 0.25]$.

**Dataset construction.** We use the fourier graphon family with $C = 8$ classes, $\rho=0.5$, and num_terms=8 (coeff_scale = 0.25 in config). Training sizes are $128, 256, 512, 1024$ and test sizes are $128, 256, 512, 1024, 2048$. The 100k node budget is applied *per class* by cycling through sizes and allocating graphs until the remaining budget is smaller than the next size. Let $\lambda$ be the large-graph budget fraction and $(1-\lambda)$ the small-graph fraction; the allocation cycles over $(128,256)$ for the small budget and $(512,1024)$ for the large budget. Train/Val split is a global 80/20 random split (not stratified by size). The test set uses 5 graphs per class per size, i.e., $8 \times 5 = 40$ graphs at each of 128/256/512/1024/2048 (total 200).

**Models and optimization.** We use DeepSets with $\phi$ and $\rho$ as 2-layer MLPs: $\phi$: [in_dim, 256, 256], $\rho$: [256, 256, 8], ReLU activations, mean pooling over nodes, no dropout. Positional encoding is eig (PE kind = eig) with $k$=32; eigenvectors of the normalized shift operator $\Delta$ are sorted and scaled by $\sqrt{n}$, and we add $10^{-4}I$ in the GPU batch eigendecomposition for stability. Training uses Adam with lr $5\times10^{-4}$, weight decay $10^{-4}$, for 100 epochs on NVIDIA RTX A6000. Tokens are precomputed once per graph (so the effective SGD batch size is 1 per update in this path. Randomness is controlled by NumPy's RNG with seed 0 for data generation and PE seed 0.

**Token discrepancy metric.** For each graph $G$ with tokens $\{t_i^{(G)}\}_{i \in V(G)} \subset \mathbb{R}^k$, define

$$\mu_G := |V(G)|^{-1} \sum_{i \in V(G)} \delta_{t_i^{(G)}}, \qquad \mu_{\mathcal{S}} := |\mathcal{S}|^{-1} \sum_{G \in \mathcal{S}} \mu_G.$$

We compute the train–test discrepancy by

$$\text{Discrepancy}_{\text{set}}(\mathcal{S}_{\text{tr}}, \mathcal{S}_{\text{te}}) := \mathsf{W}_1(\mu_{\mathcal{S}_{\text{tr}}}, \mu_{\mathcal{S}_{\text{te}}}),$$

estimated via node subsampling (without replacement) and sliced Wasserstein with random projections (specify $m$, number of projections, and repetitions).

## D.2. Learnable Proj-PE (SPE) implementation.

**SPE.** Beyond the hard projector, *Stable and Expressive Positional Encodings* (SPE) (Huang et al., 2023) can be viewed as a more expressive projector-style variant: rather than using only the hard top-$k$ projector $U_k U_k^\top$, it applies an *eigenvalue-equivariant* (basis-consistent) transformation in the spectral domain. Abstractly, a broad class of such constructions can be written as

$$t^{\text{spe}}(W) \approx U \, G(\Lambda) \, U^\top r,$$

where $G(\Lambda)$ is an eigenvalue-dependent operator (e.g., diagonal weighting or low-rank mixing in the eigenbasis) and $r$ is a probe/readout (simple pooling corresponds to fixing $r$, such as an all-ones vector). Compared with projecting onto $\text{span}(U_k)$, this eigenvalue-aware mixing can preserve finer information about the spectral distribution (e.g., soft frequency partitioning), which may help when different graphons have similar leading eigenspaces but differ in their eigenvalue profiles.

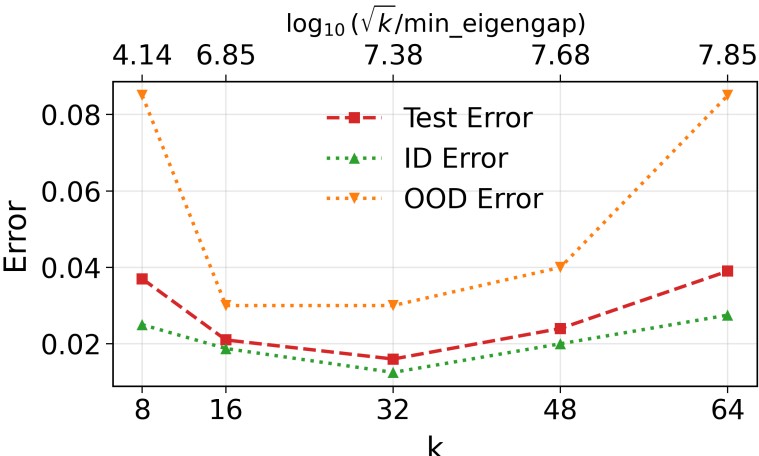

*Figure 9.* **GIN + Eig-PE: test error vs. $k$ with a message-passing backbone.** Mean over 5 independent runs. The GIN backbone exhibits the same qualitative expressivity–stability trade-off as in the DeepSets experiments: small $k$ underfits, intermediate $k$ performs best, and larger $k$ degrades again, especially on the OOD-size subset. The top axis reports $\log_{10}(\sqrt{k}/\mathrm{min\_eigengap})$, which increases with $k$ and reflects the eigengap-sensitive stability cost in Lemma 4.8.

In §5.4, we adopt an SPE-style module as the learnable Proj-PE variant. Given the normalized shift operator $\Delta \in \mathbb{R}^{n \times n}$ for each graph, we compute the top-$k$ eigendecomposition $\Delta = V \operatorname{diag}(\Lambda) V^\top$ with $V \in \mathbb{R}^{n \times k}$ and $\Lambda \in \mathbb{R}^k$. We then learn a bank of $m$ eigenvalue-dependent filters $\{\psi_j\}_{j=1}^m$ (sigmoid gates with trainable parameters applied to per-graph normalized eigenvalues) and use them to mix the top-$k$ eigenvectors. Concretely, we form $Z(\Lambda) \in \mathbb{R}^{k \times m}$ with entries $Z_{\ell j} = \psi_j(\lambda_\ell)$ and output $t_G^{\mathrm{spe}} = V Z(\Lambda) \in \mathbb{R}^{n \times m}$, optionally followed by a lightweight channel-wise MLP. All SPE parameters are trained jointly with the downstream DeepSets backbone.

# E. Additional Experiments

## E.1. Eig-PE's Effect on Generalization with GIN Backbone

We further test whether the expressivity–stability trade-off observed in Section 5.4 is specific to the DeepSets backbone. To this end, we replace the downstream predictor with GIN (Xu et al., 2019b) while keeping the same Eig-PE construction. Figure 9 shows that GIN exhibits the same qualitative pattern: small $k$ gives high error due to limited positional expressivity, intermediate $k$ achieves the best test performance, and larger $k$ degrades again, especially on the OOD-size subset.

This trend is consistent with the gap-sensitive PE-stability bound $C_{\mathrm{eig}} \propto \sqrt{k} \max_{\ell \leq k} \gamma_\ell^{-1}$ from Lemma 4.8. Increasing $k$ enriches the spectral tokens, but also exposes the model to smaller eigengaps and therefore larger eigenvector-token perturbations under sampling noise and graph variation. The GIN result suggests that this phenomenon is not an artifact of the permutation-invariant DeepSets backbone; it also appears when the predictor uses message passing.

We restrict the GIN sweep to $k \leq 64$. For larger $k$, many GIN runs become optimization-limited and occasionally collapse to near-chance accuracy. We therefore exclude those runs from this PE-stability analysis, since they reflect a separate training-stability issue rather than the eigengap-driven mechanism studied here.

## E.2. Empirical Validation of the Lipschitz Assumption on Pretrained GFMs

Our decomposition (Theorem 4.11) assumes an $L_\theta$-Lipschitz backbone. We verify this holds with bounded, stable constants on two real pretrained GFMs: **AnyGraph** (Xia & Huang, 2024) (an MLP-based mixture-of-experts model) and **GFT** (Wang et al., 2024) (a GraphSAGE-based model), evaluated across real-world datasets spanning academic, e-commerce, social, and P2P domains (2.7K–144K nodes). For a differentiable model $f$, the local Lipschitz constant at a point $x$ equals the spectral norm of its Jacobian, $\|J_f(x)\|_2 = \sup_{\|v\|=1} \|J_f(x)v\|$, and the global constant is $L = \sup_x \|J_f(x)\|_2$. We estimate $\|J_f(x)\|_2$ by sampling 30 random unit directions $v_k$ and computing $\max_k \|f(x + \epsilon v_k) - f(x)\|/\epsilon$ (a lower bound that tightens with more samples), repeated at 10 nearby points per dataset to assess stability.

For AnyGraph, the estimated constant is $L \approx 4.4$ with per-point std 0.02–0.06; for GFT, $L \approx 20$ with std $\leq 0.16$. These

constants are stable across heterogeneous domains (cross-dataset std $= 0.43$ for AnyGraph), do not grow with graph scale (Amazon-book at 144K nodes yields $L = 3.97$, lower than Cora at 2.7K), and are measured on pretrained backbones *without* fine-tuning—the hardest setting, since fine-tuning can only further regularize. The low variance across diverse inputs indicates the estimate is tight within the data distribution, supporting that the Lipschitz assumption—standard across prior graphon transfer theory (Ruiz et al., 2020; Maskey et al., 2023; Keriven et al., 2020)—holds in practice for real GFM checkpoints.

