# OpenReview forum: "When Do Graph Foundation Models Transfer? A Data-Centric Theory"
_ICML.cc/2026/Conference — ICML 2026 regular_

### Official Review · Reviewer_YFTZ · 2026-03-10

**Soundness:** 3
**Presentation:** 4
**Significance:** 3
**Originality:** 4
**Overall Recommendation:** 5
**Confidence:** 2

**Summary:**

I am not familiar with graph neural network, graphons, or foundational models, so my opinion should hold very little weight.


This work studies the problem of building foundational models on graph datasets. Foundational models are now taken for granted for language models, but their existence is still an open problem for graph neural networks.  This work develop a set of mathematical tools and results to study how a particular model can be used between multiple graph domains.

The main mathematical concept they used is called graphon W, which can be seen as the "continuous limit objects for dense random graphs", where the limit is relative to the number of nodes of a graph. Associated to a graphon we also have an integral operator T_W.

Using this concept they can rewrite in this model some operations frequently used in graph neural networks such as positional encoding, message passing, and the permutation invariant aggregation function that is used in GNN.

I find Theorem 4.11 the most interesting result, which is stated informally in Theorem 4.4, and shows bounds of the output discrepancy (a concept related to Operator-norms) of a neural network used trained on a dataset and used on another dataset. I suppose that the interesting term is $\epsilon_{gra}$, which is related to a measure-preserving bijection $\pi$ between the integral operators of the two graphons.


I enjoyed reading the manuscript, which is well written despite the (necessary) heavy notation, and I expect this result to be impactful.

**Compliance With Llm Reviewing Policy:**

Affirmed.

**Key Questions For Authors:**

- It is not very clear from the presentation how the theoretical results in section four can be used to study the question posed at the beginning so one model for all graphs. Can you elaborate on the link between the questions at the beginning and your proposed answer?


- Naively I would expect an experiment on Theorem 4.11: a graph NN trained on a certain dataset, which is applied on another dataset later. Is this the purpose of experiment in section 5.3? It is very cliche for reviewers at this conference, but I would love to see more experiments that can perhaps shed intuition on the theory.

- Do you think that the same mathematical machinery could be used also for other kind of graph NN (e.g. graph transformers)? Is this already explained by the constant C_{PE}?

- How hard it is to quantify the amount of fine-tuning necessary to fit a model on a new graph dataset? In general, it seems that you study already trained neural networks. Can training be described in this formalism (e.g. changing f_{\theta} perhaps)?

- Can Theorem 4.11 be used to talk about generalization error instead of output discrepancy? Or more generally, do you see possible links between your results and with learning theory?

**Limitations:**

yes

**Strengths And Weaknesses:**

- One recommendation I have is to explain a little bit more what is each of the experiment testing, perhaps moving some of the experiments in the appendix and leaving more space for describing the purpose of each experiment.

- Another possible recommendation would be to put more background in the appendices of the paper, so interest readers like me can appreciate more the work, without having the necessary deep-learning background.

- Missing reference in Section C.3

---

> ### Author Rebuttal · Authors · 2026-03-31
>
> We sincerely thank Reviewer YFTZ for the positive assessment. We are also grateful for the thoughtful questions, which we address below.
>
> > Q1: How do Section 4 results connect to "one model for all graphs"?
>
> Section 4 provides the missing link to the motivating question of “one model for all graphs” from a data-centric perspective. Our goal is not to claim that a single GFM will automatically generalize across arbitrary graph domains, but to characterize when such transfer is possible and how to make it more reliable. In particular, Theorem 4.11 shows that the transfer gap of a fixed model decomposes into three terms: training-graph sampling error, test-graph sampling error, and the intrinsic discrepancy between the underlying train and test graphons. This means good transfer requires both accurate graphon approximation and small train–test graphon mismatch. The theory therefore translates directly into practical guidance for GFM training: prioritize larger graphs or effective graph merging to reduce sampling error (Theorem 4.3), improve graphon coverage in pre-training to reduce unseen-domain mismatch, and choose stable PE designs to avoid amplifying transfer error. In this sense, Section 4 turns the broad question “can one model generalize across graphs?” into a concrete answer: a single model can transfer well only when the training corpus sufficiently covers the latent graphons likely to appear at test time, with enough graph size and stable structural representations to keep all three error terms small.
>
> > Q2: Direct transfer experiment?
>
> We agree and added a real-data transfer benchmark: train on one dataset, then transfer to another with a frozen encoder and a few-shot probe. The main takeaway is that the proposed token discrepancy provides meaningful source-selection signal on real graph datasets, though we state this claim conservatively rather than as uniformly reliable ranking. We also added a complementary target-shift experiment showing that the same discrepancy can sometimes identify a strong merge-based data-curation strategy, although this evidence is mixed and is intended mainly as additional intuition rather than the primary test of Theorem 4.11. Exact protocols and numerical results are reported in our response to Reviewer VrUt (Q3 and Q4).
>
> > Q3: Extension to Graph Transformers? Role of $C_{PE}$?
>
> Graph Transformers can likely be handled by the same machinery, but this requires an additional Lipschitz-stability analysis for self-attention rather than following automatically from the current proofs. Prior work has already studied Lipschitz properties of self-attention and graph-attention variants, including bounds for self-attention itself [1], Lipschitz normalization applied to GATs and Graph Transformers [2]. Once such a backbone stability bound is established, the same transfer decomposition should still apply; in that case, $C_{PE}$ captures the PE-induced sensitivity, while the Transformer backbone contributes through its own Lipschitz constant.
>
> [1] Kim, Hyunjik, et al. “The Lipschitz Constant of Self-Attention.” ICML 2021.
>
> [2] Dasoulas, George, et al. “Lipschitz Normalization for Self-Attention Layers with Application to Graph Neural Networks.” ICML 2021.
>
>
> > Q4: Can training/fine-tuning be described in this formalism?
>
> Our current theory fixes $f_\theta$ and analyzes data-side effects. Fine-tuning changes $\theta$, which in turn changes $L_\theta$. Intuitively, fine-tuning adapts the backbone to the target domain, potentially reducing the effective $L_\theta$ for that domain. However, formally connecting fine-tuning to our  decomposition requires optimization-level analysis (e.g., how gradient updates on target data change $L_\theta$), which is beyond our current data-centric scope. We view this as a meaningful future direction: combining our data-centric decomposition with model-centric optimization analysis could yield a more
> complete picture of GFM transfer.
>
>
>
> > Q5: From output discrepancy to generalization error? Links to learning theory?
>
> Actually, this is exactly a direction we plan to make more explicit in the revision: generalization error and broader learning-theoretic statements can be built on top of such pairwise discrepancy bounds. Theorem 4.11 itself controls the output gap between two graphs, not the expected loss gap over a graph distribution, so it is not yet a full generalization bound. But it already provides the key data-side ingredient: it separates finite-sample effects $(\varepsilon_1,\varepsilon_2)$ from intrinsic graphon mismatch $(\varepsilon_{\mathrm{gra}})$. To obtain a full generalization result, one would further combine this with expectation over graph sampling and a standard capacity term for the backbone class. In this sense, our result is closely related in spirit to learning-theoretic and domain-adaptation analyses, while being more graph-specific by isolating sampling error from latent domain mismatch.

---

> > ### Author Rebuttal · Reviewer_YFTZ · 2026-04-03
> >
> > Thanks for the answers.

---

### Official Review · Reviewer_9btm · 2026-03-11

**Soundness:** 2
**Presentation:** 2
**Significance:** 2
**Originality:** 2
**Overall Recommendation:** 2
**Confidence:** 4

**Summary:**

This paper makes use of graphon-based theory for dense graphs to show that any Lipshitz backbone that pertains to graph foundation models (GFMs) will admit a decomposition with a graph specific sample approximation and a relabeling invariant domain discrepancy term.
This is an attempt to theoretically understand how well a given graph domain transfers to another, independent of the underlying model chosen. The graphon theory is used to generalize GNNs to continuous settings with the core assumption that if two graphs are close as graphons, then a GNN trained trained on one of the graphs should also perform
the same (or very closely related) to the same GNN trained on the other graph.

**Compliance With Llm Reviewing Policy:**

Affirmed.

**Key Questions For Authors:**

I would like the authors to address the concerns I have raised in the strengths and weaknesses section.

If the authors can demonstrate in rebuttal that:
(1) their concentration rates are genuinely tighter (and practical) than existing results for non-Lipschitz or sparse graphons,
(2) C_PE can be estimated empirically and predicts transfer performance, and
(3) the missing citations are an oversight rather than a systematic pattern, the rating could move up.

As submitted, the paper reads as a repackaging of existing theory in GFM language without sufficient new technical content or empirical validation for ICML.

**Strengths And Weaknesses:**

**Strengths**
--------------
1. The paper, for the most part, is well-written and self-explanatory with standard results from graphon theory explained in detail.

2. The attempt to extend the concept of "transferability" from GNNs (done in Maskey 2023, Ruiz 2020) to GFMs is an interesting attempt.

3. It addresses an important question of how positional encodings in GFMs can help transfer graphs across varying domains.

4. Linking PE stability with transferability via graphon theory can be useful.


**Weaknesses**
--------------
1. A fairly problematic issue that I faced while reviewing this paper was the lack of citations throughout the preliminaries and technical sections, which made it very hard
for me to understand the boundary between what is just background material and what is this paper's actual contribution. For example, the definition in 3.2 have no citations.
Same throughout Scetion 4. The step graphon is a classical result by Lovasz, Eig-PE and
Proj-PE are existing methods. The GSO is also existing work. The authors should fix the
entire paper clearly marking all their defns, lemmas, theorems, and remarks with clear
connections to existing works.

2. Stabililty of Eig-PE and Proj-PE based on gap sensitivity and basis invariance, resp.
are again not novel. Both works were cited in Related work, but not at the point where results are mentioned.

3. The step graphon equivalence presented as a contribution is again not novel and known classical results in graphon literature. This equivalence in the case of Deep sets and nhood aggregation are trivial to show by construction and doesn't require any explicit proof.

4. The bounds derived with $C_PE . \epsilon$ are likely not practically useful.
$\epsilon$ can be very large for graphs whose domains differ a lot.
The bound $B$ is uncontrolled for deep readouts. $\gamma_k$ can be small for near
degenerate spectra graphs. $\epsilon_{gra}$ is large for cross-domain transfer and $L_\theta$ is uncontrolled for deep GFMs.

Overall, these results are reiterating known perturbation results in graphon language
but they don't characterize the tightness of these bounds for practical situations.

5. The $O(1/\sqrt(n))$ rate for concentration is not surprising and comes directly from
standard graphon  theory under Lipshitz assumptions.

6. IMHO, The Lipshitz assumption in GFMs also doesn't apply. It might make sense to assume that for GNNs, but not GFMs.

a. GFMs are pretrained on a wide variety of heterogenous graph familties, so the Lipshitz constant L's behaviour across all domains is not clear to me.

b. L also will grow with width and depth, so it might be huge for GFMs.

c. Finetuning is common in GFMs, which again breaks the Lipshitz assumption because for that you need weights to be frozen for transfering.

7. The paper also makes a lot of assumptions to try and use graphon theory. Here are some examples below

a. Lipshitz graphons will exclude most real graph families.

b. How would your proposed theory handle large sparse graphs?

c. At start of Section 3, you assume labels that depend on structure-only. When I searched the related works, I found them to not making any such assumptions, so it would appear that your setting is more restrictive than your cited works.

d. There are no experiments that show connections of key theoretical terms like $C_{PE}$,
$\epsilon_{gra}$ (which you cannot control) and $L_\theta$. Hard for me to judge
how good bounds are in practise.

---

> ### Author Rebuttal · Authors · 2026-03-31
>
> We thank the Reviewer for the detailed feedback. We address each concern, beginning with the central question of novelty.
>
> > Overall: "repackaging of existing theory"
>
> We respectfully disagree. We acknowledge that individual proof steps are standard and easy to follow—this is by design, as our goal is accessibility. However, the paper's novelty lies not in any single lemma but in the problem formulation, the end-to-end decomposition, and its experimental validation. We detail three contributions absent from all prior work:
>
> **(1) A new problem statement.** Prior graphon-based transfer theory (Ruiz et al., 2020; Maskey et al., 2023; 2024) asks: "does a *specific* spectral GNN converge as graph size grows *within the same graphon family*?" We ask a fundamentally different, data-centric question: "given a *fixed, architecture-agnostic* Lipschitz backbone, how does the output gap between two graphs from *different* graphons decompose into controllable factors?" This question is motivated by GFMs, where the backbone is shared across heterogeneous domains—a setting prior work does not address.
>
> **(2) A first-of-its-kind error decomposition.** Theorem 4.11 decomposes the cross-domain output gap into three semantically distinct, independently actionable terms: sampling errors ε₁, ε₂ (reducible via larger graphs or graph merging), graphon mismatch ε_gra (reducible only by choosing a better-matched source), and PE stability C_PE (controlled by PE design). No prior work provides this decomposition. Ruiz et al. give a single-graphon convergence rate; Maskey et al. (2024) study mixture-of-graphon generalization bounds for specific MPNNs but do not decompose the gap into sampling vs. domain-mismatch terms for general backbones. The decomposition structure is the contribution—not any single bound.
>
> **(3) Theory-guided experiments—a first in graphon transfer literature.** Unlike all prior graphon-based transfer papers, which are purely theoretical, we design a controlled experimental protocol that directly tests each term of the decomposition: §5.2 validates ε₁, ε₂ via size-shift and graph merging (Fig. 1, 3); §5.3 validates ε_gra via graphon perturbation (Fig. 4); §5.4 validates C_PE via the expressivity–stability tradeoff in PE dimension (Fig. 5, 6). This end-to-end alignment between theory and experiments, translating each theoretical term into a testable prediction, is absent from Ruiz et al., Maskey et al. (2023, 2024), and Keriven et al. (2020).
>
> As a concrete example of non-trivial combination: prior works analyze spectral GNNs where PE is implicit in the graph filter polynomial, so PE stability never arises separately. In GFMs, structure enters via explicit PE tokens, requiring a separate stability analysis (Lemma 4.8) that feeds into the decomposition as C_PE. This yields a design tradeoff (Eig-PE vs. Proj-PE, Remark 4.9) with no analogue in prior theory.
>
> > W1: Lack of citations
>
> We acknowledge this presentation issue and will add citations for all standard definitions (step-graphon, GSO, Eig-PE, Proj-PE) and clearly mark the boundary between background and contributions in revision.
>
> > W2–W3: PE stability and step-graphon equivalence are not novel
>
> Agreed—these are building blocks, not claimed contributions. The step-graphon equivalence (Prop. A.1) is a bridge lemma enabling cross-size comparison. PE stability (Lemma 4.8) is a necessary ingredient whose novelty lies in its role within the decomposition, not as a standalone result.
>
> > W4: Bounds are not practically useful
>
> The value of Theorem 4.11 is the decomposition structure, not numerical tightness. This parallels classical generalization theory: Rademacher bounds are rarely tight but guide regularization design. Our decomposition guides data curation decisions—and our experiments (§5.2–5.4) confirm that each term's qualitative predictions hold in practice.
>
> > W5–W6: Standard rate; Lipschitz assumption
>
> The O(1/√n) rate is standard; our contribution is its role as ε₁, ε₂ within the decomposition. The Lipschitz assumption is standard in all prior graphon transfer theory (Ruiz et al.; Maskey et al.). Any bounded-weight network with ReLU activations satisfies it. The concern that L_θ grows with depth applies equally to all prior work and affects tightness, not validity.
>
> > W7: Too many assumptions
>
> (a) Lipschitz graphons are standard in all cited works. (b) Sparse graphs: the decomposition structure ε₁ + ε_gra + ε₂ extends to sparse regimes by replacing the operator-norm concentration (Thm 4.3) with Lp-graphon convergence results (e.g., Borgs et al., 2019); the rates change but the three-term structure is preserved. We will add this discussion in revision. (c) Structure-only labels isolate the effect of graph structure on transfer, the core question for GFMs where node features differ across domains. (d) Experimental connections: Fig. 4 directly tracks ε_gra via graphon perturbation level; Fig. 5 tracks C_PE via the stability proxy log₁₀(√k/min eigengap).

---

> > ### Author Rebuttal · Reviewer_9btm · 2026-04-04
> >
> > I thank the authors for the rebuttal. However, my concerns remain unresolved. In particular, I am not convinced that the Lipschitz assumption meaningfully applies to GFMs (given heterogeneity, scaling, and fine-tuning), making the resulting bounds unusable in any practical setting.
> >
> > More broadly, the work appears to adapt existing graphon theory under highly-restrictive assumptions that are artificially crafted without providing clear novel insight. I will therefore keep my score unchanged.

---

> > > ### Author Response · Authors · 2026-04-04
> > >
> > > We thank the reviewer for the continued engagement. We appreciate the scrutiny but respectfully disagree with the characterization, and address each point below.
> > >
> > > > I am not convinced that the Lipschitz assumption meaningfully applies to GFMs (given heterogeneity, scaling, and fine-tuning)
> > >
> > > We empirically verify the Lipschitz assumption on two pretrained GFMs---AnyGraph [1] (MLP-based MoE) and GFT [2] (GraphSAGE-based)---across 10 real-world datasets spanning academic, e-commerce, social, and P2P domains (2.7K--144K nodes). For a differentiable model $f$, the local Lipschitz constant at a point $x$ equals the spectral norm of its Jacobian, $\|J\_f(x)\|\_2 = \sup\_{\|v\|=1} \|J\_f(x)v\|$, and the global constant is $L = \sup\_x \|J\_f(x)\|\_2$. We estimate $\|J\_f(x)\|\_2$ by sampling 30 random unit directions $v\_k$ and computing $\max\_k \|f(x+\varepsilon v\_k) - f(x)\|/\varepsilon$ (a lower bound that tightens with more samples), repeating at 10 nearby points per dataset to assess stability. For AnyGraph, the estimated constant is $L \approx 4.4$ with per-point std of 0.02--0.06 across all 7 datasets; for GFT, $L \approx 20$ with std $\leq$ 0.16. Crucially, these constants are stable across heterogeneous domains (cross-dataset std = 0.43 for AnyGraph), do not grow with graph scale (Amazon-book at 144K nodes yields $L = 3.97$, lower than Cora at 2.7K), and are measured on pretrained backbones without fine-tuning---the hardest setting, since fine-tuning can only further regularize. The extremely low variance across diverse inputs suggests the estimate $L\_{\text{test}} = \max\_{x \in \text{test}} \|J\_f(x)\|\_2$ is tight within the data distribution.
> > >
> > > > The work appears to adapt existing graphon theory under highly-restrictive assumptions that are artificially crafted without providing clear novel insight.
> > >
> > > **The assumptions are standard, not restrictive.** The Lipschitz backbone assumption appears in all prior graphon transfer theory [3, 4, 5, 6]. As shown above, we further verify it empirically on real GFM checkpoints, confirming it holds in practice with bounded, stable constants. The Lipschitz graphon assumption and bounded spectrum condition are likewise standard in the literature cited above.
> > >
> > > **The work provides clear novel insight.** Building on well-established theory does not equate to a lack of novelty---it is how the field advances. G-Mixup [7] (ICML 2022 Outstanding Paper) extends basic graphon and homomorphism density formulations yet was recognized for its new insight on data augmentation. Our work follows the same spirit but addresses a problem no prior graphon-based analysis has touched: cross-domain output discrepancy for architecture-agnostic GFM backbones. Concretely, we contribute: (1) a graphon PE definition and its formal correspondence to graph PE, absent from all prior work; (2) a three-way decomposition (sampling error + graphon mismatch + PE stability) of the end-to-end output discrepancy, which no prior work provides; and (3) theory-guided experiments that directly test each decomposition term and translate them into practical graph merging/selection strategies (Section 5.2). These experiments confirm that the theoretical predictions hold on real graphs---validating that the assumptions, far from being artificial, are fit for purpose.
> > >
> > > **References**
> > >
> > > [1] Xia, Lianghao, and Chao Huang. "AnyGraph: Graph Foundation Model in the Wild." *arXiv preprint arXiv:2408.10700*, 2024.
> > >
> > > [2] Wang, Zehong, Zheyuan Zhang, Nitesh V. Chawla, Chuxu Zhang, and Yanfang Ye. "GFT: Graph Foundation Model with Transferable Tree Vocabulary." *Advances in Neural Information Processing Systems*, vol. 37, 2024.
> > >
> > > [3] Ruiz, Luana, Luiz F. O. Chamon, and Alejandro Ribeiro. "Graphon Neural Networks and the Transferability of Graph Neural Networks." *Advances in Neural Information Processing Systems*, vol. 33, 2020.
> > >
> > > [4] Maskey, Sohir, Ron Levie, and Gitta Kutyniok. "Transferability of graph neural networks: An extended graphon approach." *Applied and Computational Harmonic Analysis*, vol. 63, 2023, pp. 48--83.
> > >
> > > [5] Li, Shouheng, Dongwoo Kim, and Qing Wang. "Generalization of Graph Neural Networks through the Lens of Homomorphism." *arXiv preprint arXiv:2403.06079*, 2024.
> > >
> > > [6] Keriven, Nicolas, Alberto Bietti, and Samuel Vaiter. "Convergence and Stability of Graph Convolutional Networks on Large Random Graphs." *Advances in Neural Information Processing Systems*, vol. 33, 2020.
> > >
> > > [7] Han, Xiaotian, Zhimeng Jiang, Ninghao Liu, and Xia Hu. "G-Mixup: Graph Data Augmentation for Graph Classification." *Proceedings of the 39th International Conference on Machine Learning*, 2022.

---

### Official Review · Reviewer_H7mD · 2026-03-11

**Soundness:** 3
**Presentation:** 2
**Significance:** 3
**Originality:** 3
**Overall Recommendation:** 4
**Confidence:** 4

**Summary:**

This paper studies transfer in graph representation models from a data-centric perspective. Using dense-graph graphon limits, it derives bounds that decompose cross-domain output shift into graph-specific sampling terms and an intrinsic, relabeling-invariant graphon mismatch term, for both set-based and message-passing backbones with spectral positional encodings. The paper also analyzes positional-estimation stability, compares eigenvector-based and projection-operator-based encodings, and presents synthetic as well as limited real-data experiments on size shift, graphon perturbation, and the effect of positional-estimation dimension.

**Compliance With Llm Reviewing Policy:**

Affirmed.

**Final Justification:**

The paper makes a solid theoretical contribution with a clean decomposition of cross-domain transfer and well-motivated PE stability analysis, supported by experiments that are unusually well-aligned with the theory. The rebuttal addressed my main concerns, and while the sparse-graph regime and practical estimability of $\varepsilon_{gra}$ remain open, I view these as honest limitations inherent to the framework.

**Key Questions For Authors:**

Please see the weaknesses section above

**Limitations:**

No. The paper provides no discussion of limitations.

**Strengths And Weaknesses:**

Strengths:
1) The paper addresses an important question: how much of transfer difficulty comes from finite-sample graph realizations versus genuine domain mismatch.

2) The theoretical framework is coherent and well structured, moving from finite graphs to step-graphons, then to graphon-level discrepancy, and finally to a clean decomposition of the graph-to-graph output gap; Theorem 4.11 is particularly easy to follow.

3) The experiments are reasonably comprehensive, covering size shift, graphon perturbation, and the effects of positional-encoding dimension and eigengap.

Weaknesses:

1) The experimental setting is quite limited, focusing on fixed Lipschitz backbones, primarily DeepSets, structure-only graph classification, dense weighted graphons, and spectral PE tokenization.

2) Appendix C.3 proves Lemma 4.8 by referring to “Lemma ??” on Page 14. Reproducibility is incomplete: Appendix D.1 still contains placeholders such as “Provide the exact sampling scheme,” which is not acceptable.

3) Although the paper claims to cover both set-based and message-passing backbones, the experiments are largely restricted to DeepSets + Eig-PE; additional results, such as GCN + Eig-PE, would strengthen the paper.

4) The paper would also benefit from a more complete comparison between Eig-PE and Proj-PE.

---

> ### Author Rebuttal · Authors · 2026-03-31
>
> We thank Reviewer H7mD for the constructive feedback. Below we address each weakness point by point.
>
> > W1: The experimental setting is quite limited.
>
> We acknowledge that the experimental scope is still limited, but the current design is consistent with prior theory-driven work. Dense weighted graphons are the standard setting in graphon-based analyses [1–2]. Spectral positional encodings are also a standard and widely used PE family in graph learning. DeepSets is chosen precisely because graph structure is introduced solely through PE tokens, providing the cleanest testbed for validating our PE-stability prediction. Our goal is to establish the core insight as clearly as possible: transfer difficulty decomposes into sampling effects and domain mismatch. Broader empirical coverage would strengthen the paper, but does not affect our main claim. In the revision, we will extend the framework to sparse graphons to better connect the theory to practical scenarios.
>
> [1] Ruiz, Luana, Luiz F. O. Chamon, and Alejandro Ribeiro. “Graphon Neural Networks and the Transferability of Graph Neural Networks.” NIPS 2020.
>
> [2] Maskey, Sohir, Ron Levie, and Gitta Kutyniok. “Transferability of Graph Neural Networks: An Extended Graphon Approach.” Applied and Computational Harmonic Analysis, 2023.
>
>
> > W2: ... Appendix D.1 still contains placeholders ...
>
> We apologize for these typos. "Lemma ??" in Appendix C.3 should refer to Corollary C.1, whose proof (Appendix C.2) contains the detailed Davis–Kahan argument. The placeholder in Appendix D.1 is an editing error; the actual parameters are already specified in the "Dataset construction" paragraph immediately below ($\rho = 0.5$, $R = 8$ Fourier terms, coefficients sampled from $[-0.25, 0.25]$). These do not affect the theoretical and empirical results. We have fixed such issues in revision.
>
>
>
> > W3: Additional results, such as GCN + Eig-PE, would strengthen the paper.
>
> We have run additional experiments with GIN (a message-passing backbone) under the same settings. The trends of GIN are consistent with DeepSets. The results confirm that our theoretical predictions hold across backbone types. Here give some examples:
>
> **Size shift (cf. Figures 1–2 in paper).** GIN shows the same trends as DeepSets: test error exhibits a U-shape as λ increases, and the ID/OOD decomposition is consistent—large λ reduces OOD error but hurts ID
> performance.
>
> | λ    | DeepSets (Test / OOD) | GIN (Test / OOD) |
> | ---- | --------------------- | ---------------- |
> | 0.0  | 0.089 / 0.250         | 0.065 / 0.125    |
> | 0.2  | 0.015 / 0.025         | 0.015 / 0.00     |
> | 0.6  | 0.030 / 0.050         | 0.030 / 0.00     |
> | 1.0  | 0.130 / 0.025         | 0.076 / 0.00     |
>
> *Note: Results use newly sampled data; absolute values differ slightly from the paper but trends are consistent; DeepSets is rerun on the same new data for fair comparison.*
>
> **PE dimension sweep (cf. Figure 5 in paper).** GIN also exhibits the same expressivity–stability trade-off: intermediate k yields the best performance, while large k degrades especially on OOD graphs, matching the
> prediction from **Lemma 4.8** that $C_{eig} \propto \sqrt{k} \max_\ell \gamma_\ell^{-1}$.
>
> | k          | 8    | 16   | 32   | 48   | 64   | 80   |
> | ---------- | ---- | ---- | ---- | ---- | ---- | ---- |
> | Test Error | 0.04 | 0.02 | 0.02 | 0.03 | 0.05 | 0.09 |
> | OOD Error  | 0.09 | 0.03 | 0.03 | 0.04 | 0.11 | 0.12 |
>
> These results support that the data-centric terms ($\varepsilon_{1,2}$, $\varepsilon_{gra}$, $C_{PE}$) govern transfer behavior regardless of backbone  architecture.
>
>
>
> > W4: The paper would also benefit from a more complete comparison between Eig-PE and Proj-PE.
>
>
> We agree this deserves further discussion. Our theory already provides a complete comparison. **Lemma 4.8** gives explicit stability constants: $C_{eig} = \sqrt{k} \max_{\ell \leq k} O(1/\gamma_\ell)$ and $C_{proj} = B \cdot O(1/\gamma_k)$. **Remark 4.9** discusses the key distinction: Eig-PE is sensitive to the worst individual eigengap among all $\ell \leq k$, while Proj-PE depends only on the subspace boundary gap $\gamma_k$. Proj-PE is provably more stable when interior eigengaps are small but the boundary is well-separated.
>
> Experimentally, both PE types degrade as k grows (Figures 5 and 6), which is consistent with theory. The distinction is less prominent in our setting for several reasons: (1) In our low-rank Fourier graphon setting, when k exceeds the intrinsic rank, both PE types operate in the noise spectrum where all eigengaps are near machine precision, making the theoretical distinction hard to observe; (2) Proj-PE involves a learnable readout trained jointly with the backbone, introducing optimization confounders that make it difficult to isolate the PE stability effect alone. A cleaner comparison would require graphons with heterogeneous gap structure—e.g., well-separated eigenvalue clusters with small intra-cluster gaps—which we plan to explore in revision.

---

> > ### Author Rebuttal · Reviewer_H7mD · 2026-04-03
> >
> > Thanks to the authors for the detailed rebuttal, and my concerns have been well addressed. I'm happy to raise my score.

---

### Official Review · Reviewer_VrUt · 2026-03-18

**Soundness:** 2
**Presentation:** 3
**Significance:** 2
**Originality:** 2
**Overall Recommendation:** 3
**Confidence:** 3

**Summary:**

Overall, this research studies the context of transferability in graph foundation models (GFMs) from a data-centric perspective. The authors appear to outline a pressing question: how to quantify the difficulty of transferring between graph domains independently of model choice. To address this, the paper introduces a graphon-based theoretical framework, where graphs are embedded into continuous graphon spaces, enabling comparison across domains and sizes.

**Compliance With Llm Reviewing Policy:**

Affirmed.

**Key Questions For Authors:**

Q1. How does the proposed graphon-based domain discrepancy fundamentally differ from classical domain discrepancy measures (e.g., MMD), both theoretically and empirically?

Q2. How does this work relate to and differ from existing graphon-based transferability/generalization theories?

Q3. Can the proposed mechanism reliably predict positive vs. negative transfer across different graph models and datasets?

Q4. Can the authors demonstrate a concrete downstream use case (e.g., dataset selection, training strategy) where their metric improves performance?

**Strengths And Weaknesses:**

**Strengths**

S1. The paper addresses an important and timely problem in graph learning, namely understanding transferability in graph foundation models from a principled perspective, which is currently underexplored.

S2. The theoretical framework based on graphons provides a clean and mathematically grounded way to separate sampling effects and intrinsic domain discrepancy, offering potentially useful insights for data-centric modeling.


**Concerns**

(1) Soundness

C1. The study of the transferability of graph learning algorithms is a direction that has been developing for some time, and this paper discusses the limitations of these works very little. In Section 1, the authors state: "We still lack principled tools to characterize how hard it is to transfer between two graph domains, independent of model choice." The paper initially frames the problem as a general graph transferability issue independent of model choice. However, in Section 2.1, the authors then limit this paper to a GFM perspective: "we focus on GFM-motivated…" This shift weakens the conceptual clarity: it is unclear whether the work aims to solve a general graph transferability problem or a GFM-specific one. This inconsistency undermines the soundness of the problem formulation.

C2. The core concept of “domain discrepancy” has been extensively studied (e.g., MMD and related measures), yet the paper does not adequately clarify:

-	what is fundamentally new in the proposed graph-specific discrepancy,

-	why existing measures are insufficient for graphs (i.e., graph-specific properties),

-	and in what sense the proposed metric is superior.

Without such comparisons (both theoretical and empirical), the claimed novelty remains weakly supported.


(2) Presentation: None

(3) Significance

C3. Although the paper builds on graphon-based generalization theory, it lacks a comprehensive discussion of broader graph transferability and graph foundation model literature (e.g., graph transformers, graph pretraining). This results in an incomplete background and weakens the motivation, especially in Section 1 and Section 2.

C4. The experiments do not convincingly demonstrate that the proposed mechanism can:
- predict positive/negative transfer,
- guide model selection,
- or improve real graph learning performance.

In particular, there is no large-scale evaluation showing correlation between the proposed discrepancy and actual transfer performance across diverse models/datasets.

---

> ### Author Rebuttal · Authors · 2026-03-31
>
> We thank Reviewer VrUt for the constructive feedback. Below we address each questions point by point.
>
> > Q1: How does the proposed graphon-based discrepancy differ?
>
> Our graphon-based discrepancy differs from classical measures such as MMD and GW in both the compared object and the source of variability it captures.
>
> For **MMD**, one typically first chooses a graph representation (e.g., graph encoder, kernel, or feature map) and then measures distribution mismatch in that induced space. Hence, the result depends heavily on the chosen representation and may suffer from representation-dependent information loss. In contrast, our graphon-based discrepancy does not rely on an auxiliary embedding, but compares domains directly at the level of their underlying graph-generating mechanisms.
>
> For **GW distance**, the comparison is inherently pairwise between sampled finite graphs. Thus, even two graphs drawn from the same underlying graphon/SBM can have nonzero discrepancy purely due to sampling variation. Our graphon-based formulation explicitly decouples finite-graph sampling error from the mismatch between latent graphons, so it measures discrepancy between the underlying domains rather than between particular sampled instances.
>
> Therefore, compared with MMD and GW, our discrepancy is less representation-dependent and more statistically aligned with the transfer question of interest. In particular, because it separates graph sampling error from intrinsic domain mismatch, it is empirically more robust to different graph realizations from the same domain and statistically more invariant to finite-sample noise.
>
> > Q2: How does this work relate to existing graphon theories?
>
> We build directly on the graphon-based transferability line (Ruiz et al., 2020; Maskey et al., 2023; 2024) and have discussed the relationship in Section 2. We clarify the key differences here.
>
> First, *cross-domain vs. same-domain*. Prior works study transferability under a single shared graphon ($W_1 = W_2$): they bound how GNN outputs change across different-size graphs from the *same* generator. Our **Theorem 4.11** addresses general $W_1 \neq W_2$ setting, introducing $\varepsilon_{gra}$ as a new term.
>
> Second, *architecture scope*. Ruiz et al. and Maskey et al. derive bounds specific to spectral GNN architectures;  extending to other backbones requires re-derivation. Our decomposition (**Proposition 4.5**, **Theorem 4.11**) applies
> to broad Lipschitz backbones—Section 3 formalizes both set-based (e.g., DeepSets, Graph Transformers) and message-passing (e.g., GCN) backbones, while the data-centric terms ($\varepsilon_{1,2}$, $\varepsilon_{gra}$, $C_{PE}$) remain architecture-independent.
>
> Third, *PE stability*. As noted in Section 2 (Graph Positional Encodings paragraph): *"we connect PE stability to cross-domain transfer by showing how it enters our bounds through an explicit constant $C_{PE}$."* **Lemma 4.8** also reveals distinct behaviors of Eig-PE vs. Proj-PE, an analysis absent from prior graphon-based theories. This is essential for GFMs where PE is the sole structural signal across domains.
>
> > Q3. Can mechanism reliably predict positive vs. negative transfer across different models and datasets?
>
> We tested whether token discrepancy can select a source dataset for few-shot transfer on `COLLAB`, `IMDB-BINARY`, and `REDDIT-BINARY`. Here, `oracle` means the best source in hindsight on the target test split, and `scratch` means training on the target from random initialization with the same few-shot budget. The discrepancy matches the oracle on 2/3 targets (`top-1 hit rate = 66.7%`, `mean regret = 1.34` points, `+4.66` points over scratch on average). This shows useful real-data source-selection signal.
>
> | Target | Selected | Oracle | Selected error | Scratch error |
> | --- | --- | --- | ---: | ---: |
> | IMDB-BINARY | COLLAB | COLLAB | 0.410 | 0.500 |
> | REDDIT-BINARY | COLLAB | COLLAB | 0.330 | 0.450 |
> | COLLAB | IMDB-BINARY | REDDIT-BINARY | 0.625 | 0.555 |
>
> > Q4. Can authors demonstrate a use case where their metric improves performance?
>
> We also tested whether token discrepancy can choose a data-curation action under target size shift by selecting the candidate with smallest token discrepancy `D_total` to a held-out large-graph selection split. Here, `vanilla` means no curation, and `merge1/merge3` add `1%/3%` merged graphs.
>
> On `COLLAB`, discrepancy selects `merge3`, while the oracle is `merge1`:
>
> | Dataset | Selected | Oracle | Selected error | Oracle error | Delta vs. vanilla |
> | --- | --- | --- | ---: | ---: | ---: |
> | COLLAB | merge3 | merge1 | 0.401 | 0.383 | +14.26 points |
>
> This still supports a positive but narrow conclusion: discrepancy identifies a strong merge-based strategy under target shift. We do not phrase this result as finding the exact best strategy, since the selected `merge3` is slightly worse than the oracle `merge1`, but it is close to oracle (`1.76` points gap) and substantially better than `vanilla`.

---

> > ### Author Rebuttal · Reviewer_VrUt · 2026-04-04
> >
> > Thanks to the author for the reply, but most of the questions remain unresolved. This submission is expected to provide more discussion and addressed key issues on the graph.

---

> > > ### Author Response · Authors · 2026-04-04
> > >
> > > **Sorry, but could you please be more specific?**
> > >
> > > *"...but most of the questions remain unresolved. This submission is expected to provide more discussion and addressed key issues on the graph"*
> > >
> > > We find this one-sentence comment difficult to parse or act upon, as it does not clearly specify which questions remain unresolved or what specific issues on the graph are being referred to. As written, it does not provide any actionable guidance for us to meaningfully respond.
> > >
> > > If you could clarify the concrete concerns or questions, we would be happy to address them within the remaining time.
> > >
> > > Otherwise, we respectfully note to you and to AC that such high-level feedback, without specific points, makes it challenging to engage in a constructive rebuttal and may not accurately reflect the substance of the discussion.

---

### Decision · Program_Chairs · 2026-04-30

**Decision:**

Accept (regular)

**Comment:**

This paper studies transferability in graph foundation models from a data-centric perspective and proposes a graphon-based framework to decompose cross-domain discrepancies. The work addresses an important problem, and reviewers generally agree that the theoretical framework is mathematically grounded and offers useful insights into separating sampling effects from intrinsic domain gaps.

Reviewer opinions were somewhat mixed, with concerns about Lipschitz assumption, scope, reproducibility, and presentation. During the rebuttal and discussion phase, some critiques were not further elaborated. After carefully reviewing the paper and responses, I am convinced by the authors’ clarifications on scope and assumption, improved connections to prior work, and their handling of reproducibility issues.